# Zebrafish do not have a calprotectin ortholog

Kona N. Orlandi [1,2], Michael J. Harms [1,3]*

**1** Institute of Molecular Biology, University of Oregon, Eugene, Oregon, United States of America,
**2** Department of Biology, University of Oregon, Eugene, Oregon, United States of America, **3** Department of Chemistry and Biochemistry, University of Oregon, Eugene, Oregon, United States of America

* harms@uoregon.edu

## Abstract

The protein heterodimer calprotectin and its component proteins, S100A8 and S100A9, play important antibacterial and pro-inflammatory roles in the mammalian innate immune response. Gaining mechanistic insights into the regulation and biological function of calprotectin will help facilitate patient diagnostics and therapy for inflammation and further our understanding of the host-microbe interface. Recent literature has identified zebrafish s100a10b as zebrafish calprotectin based on sequence similarity, genomic context, and transcriptional upregulation during the immune response to bacterial infections. The field would benefit from expanding the breadth of calprotectin studies into a zebrafish innate immunity model. Here, we carefully evaluated the possibility that zebrafish possess a calprotectin ortholog or a paralog that convergently evolved similar function. Using careful bioinformatics approaches, we found that zebrafish do not have an ortholog of either mammalian S100A8 or S100A9. To look for paralogs with convergent function, we identified four zebrafish s100 proteins—including s100a10b—that are expressed in immune cells and upregulated during the immune response. We recombinantly expressed and purified these proteins and measured their antimicrobial activity. None of the zebrafish proteins exhibited activity comparable to mammalian calprotectin. We also generated structural models of all homodimers and heterodimers of all annotated zebrafish *s100* genes. None of these complexes were predicted to have an antimicrobial transition metal binding site equivalent to calprotectin. Finally, we measured the ability of our four purified zebrafish s100 proteins to activate inflammation via Toll-like receptor 4, a key feature of human S100A9; none of the proteins activated the receptor. Our work demonstrates conclusively that zebrafish have no ortholog of calprotectin and suggests that similar proteins have not convergently evolved analogous functions.

## Introduction

Calprotectin plays critical roles in mammalian innate immunity [1]. Upon release from immune cells after damage or during an immune response, calprotectin exerts

**Data availability statement:** All relevant data are within the paper and its Supporting Information files.

**Funding:** This work was funded by National Institute of General Medical Sciences R01-GM146114 (MJH), National Institute of General Medical Sciences 5T32GM007759-42 (KNO), a Molecular Biophysics and Biochemistry Training Program at University of Oregon Supported Position (KNO), a Raymond-Stevens Fellowship (KNO), and a Jeffrey McKnight Memorial Fellowship (KNO). The funders had no role in study design, data collection and interpretation, or the decision to submit the work for publication.

**Competing interests:** The authors have declared that no competing interests exist.

antimicrobial activity by sequestering transition metals essential for microbial growth in the extracellular matrix, also known as nutritional immunity [2–11]. The calprotectin protein exists as both a heterodimer and heterotetramer complex formed by two calcium binding proteins, S100A8 and S100A9. These proteins also form homodimers, allowing S100A8 and S100A9 to perform biological functions distinct from calprotectin; however, their individual roles are difficult to distinguish [12–20]. Extracellular dimeric calprotectin and S100A8 and S100A9 homodimers are thought to amplify the inflammatory response by activating Toll-like receptor 4 (TLR4) and the Receptor for Advanced Glycation End-products (RAGE), promoting a positive feedback loop of cytokine expression and immune cell migration [15,21]. Because of its high concentration and many roles in innate immunity, calprotectin has become a well-validated, non-invasive biomarker of inflammation [22].

Given the importance of calprotectin, there is interest in developing new models to study its function. One attractive model is the zebrafish (*Danio rerio*), which is increasingly being used to understand the molecular mechanisms of immune functions [23,24]. As vertebrates, zebrafish share much of their physiology and molecular components with humans. They also have exceptional experimental advantages: well-established genetic tools, optically transparent larvae (making it possible to visualize tagged molecules and microbes in real-time in live fish), and rapid generation times [25]. They are particularly useful for studying innate immunity because they rely solely on innate immune responses until 4–6 weeks post-fertilization, when their adaptive immune system is morphologically and functionally mature [23,26–28].

Recently, the zebrafish protein s100a10b was identified as "calprotectin" [29,30]. This identification was based on genomic context and sequence similarity to human S100A8. From transcriptional analyses, it was proposed that this protein plays a role in the zebrafish immune response to *Vibrio cholerae* and *Escherichia coli* infections [29,30]. Further, there are commercial ELISA antibodies marketed as targeting "fish calprotectin," implying that this protein can be found in zebrafish.

Despite the power of the zebrafish model system, it can be challenging to map zebrafish biology to human biology. Over 400 million years of evolution have allowed the divergence, emergence, and loss of proteins and protein functions between these species, often making the comparison difficult. One of the most important considerations is whether the genes being compared between species are, in fact, the same genes. Are they the result of speciation (orthologs) which often have very similar functions, or did they arise by gene duplication (paralogs) which often have very different functions?

Establishing gene orthology is particularly challenging for S100 proteins, as they form the largest subgroup within the superfamily of proteins carrying the $Ca^{2+}$-binding EF-hand motif. Humans have 24 *S100* genes [31,32]; zebrafish have 14 [33,34]. Many of these *S100* genes are in dense blocks of tandem repeats. There is no annotated *S100A8* or *S100A9* ortholog in the zebrafish genome; however, there are several zebrafish *s100* genes in a similar genomic location to that of human *S100A8* and *S100A9*.

We set out to find phylogenetic, biochemical, or biological evidence of calprotectin in zebrafish s100 proteins. Through a careful review of existing phylogenetic

literature, we confirm that fish do not have a calprotectin ortholog: both *S100A8* and *S100A9* evolved in mammals 250 million years after the divergence of tetrapods and ray-finned fishes. We support this phylogenetic result through a comparative synteny analysis of *S100* genes in zebrafish and human genomes. We also investigate the possibility that fish convergently evolved a calprotectin-like *s100* paralog. We used published single-cell RNAseq data to identify zebrafish s100 proteins expressed in immune cells. We then recombinantly expressed and purified four of these proteins—including zebrafish s100a10b, the protein previously identified as fish calprotectin in the literature—and experimentally tested their antimicrobial and pro-inflammatory activities. None of the proteins give measurable activity under the conditions of our assays. We conducted a computational structural analysis based on models of all possible zebrafish s100 homodimer and heterodimers to evaluate their similarity to calprotectin. We determined that most complexes align well with the calprotectin structure, but none are likely to form a strong antimicrobial metal chelating site like calprotectin's hexa-histidine site. We could not draw conclusions about pro-inflammatory capabilities from this computational analysis because the structural basis for calprotectin's pro-inflammatory activity, as well as zebrafish receptors that may mediate analogous immune responses, are still under investigation.

We conclude that zebrafish have neither a vertically inherited ortholog of calprotectin, nor an obvious paralog that convergently evolved similar function. Our results highlight the hazard of relying solely on sequence similarity and genomic placement to identify genes and demonstrate the importance of an explicitly evolutionary lens with careful functional analyses when mapping results from model organisms to human biology.

## Results

### Zebrafish *s100s* are distantly related to human *S100A8* and *S100A9*

We started by looking for phylogenetic evidence that fish have a protein orthologous to mammalian S100A8 or S100A9. Orthologous proteins arise by speciation and are thus the same gene in the species being compared. In contrast, paralogous proteins arise by gene duplication and often exhibit gain or loss of function from the ancestral state [35], establishing themselves as new proteins. Fig 1A summarizes the evolutionary history of S100s. This tree was built referencing several published phylogenetic analyses of the family, including two from our group [33,36–38]. The phylogeny at the top shows the current best estimate of the *S100* gene tree; the phylogeny on the left shows the evolutionary history of bony vertebrates. Each circle denotes the *S100* gene observed in at least one member of the taxonomic group on the left.

This evolutionary tree indicates that *S100A8* and *S100A9* evolved by gene duplication from a single gene in the ancestor of amniotes (shown in orange). Reptiles and birds preserve a single gene (*MRP-126*), while mammals expanded it into three proteins (*S100A8, S100A9* and *S100A12*). The reptile/bird protein MRP-126 is the earliest diverging protein known to exhibit nutritional immunity and/or Toll-like receptor 4 activation in functional assays [37,39]. These observations indicate that calprotectin evolved in amniotes ~320 million years ago.

The closest evolutionary relatives of *S100A8* and *S100A9* arose from duplications of a tetrapod-specific *S100* gene. In contrast, zebrafish *s100a10b* (the putative zebrafish calprotectin) is a member of one of the earliest *S100* gene subfamilies to evolve, with orthologs present in jawless fishes and the co-orthologous genes *S100A10* and *S100A11* in tetrapods. *S100a10b* thus diverged from the lineage that led to mammalian *S100A8* and *S100A9* at least 563 million years ago, in the last common ancestor of humans and lampreys. Further, after this speciation event, there were at least two more gene duplications on the lineage leading to *S100A8* and *S100A9*. *S100a10b* is therefore a different gene than *S100A8* or *S100A9*.

### Chromosome placement indicates a shared origin but complicated evolution of S100s in humans and zebrafish

To cross-validate the lack of evidence for vertical inheritance from published phylogenies, we used syntenic analysis to identify zebrafish *s100* genes in a similar genomic location to human *S100A8* and *S100A9*. We used ENSEMBL to identify

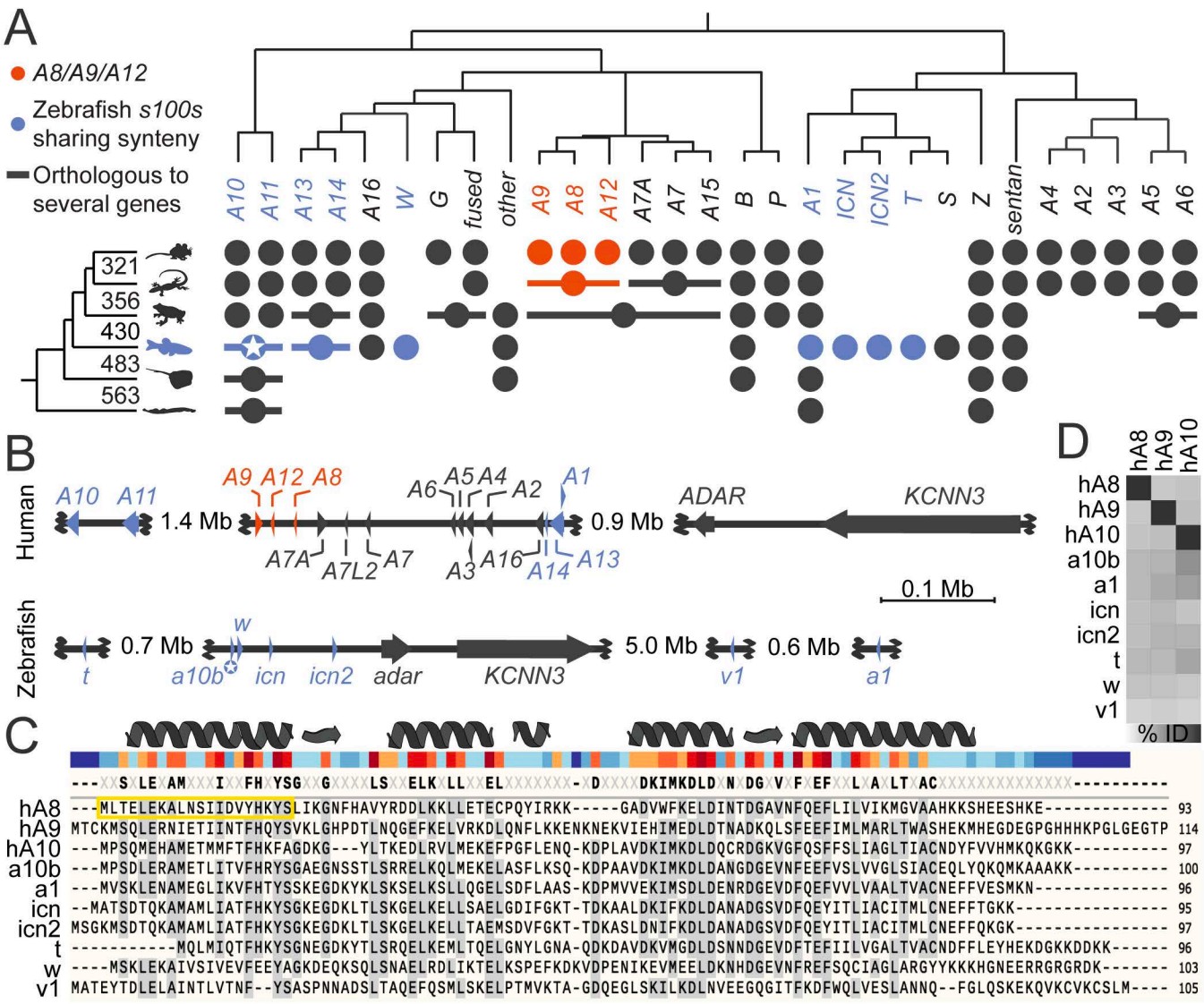

**Fig 1. Phylogenetic analyses reveal there is no calprotectin ortholog outside of amniotes. (A)** *S100* gene tree adapted from Wheeler et al., 2016 shows the evolutionary relationships determined for S100s across vertebrates. The phylogeny on the left shows the relationships between species, with branch point times noted in millions of years ago; the phylogeny on the top shows the estimated *S100* gene tree. Circles denote *S100* genes from the phylogeny at the top found in at least one member of the taxonomic group from the left. A horizontal line through a circle indicates a single gene that is co-orthologous to multiple *S100* genes found in mammals. *S100A8*, *S100A9*, and *S100A12* form a clade specific to amniotes (orange). Zebrafish genes in the syntenic region shown in panel B are shown in blue. Zebrafish *s100a10b*, which has been treated as a calprotectin ortholog, is denoted with a star. **(B)** Syntenic regions of human chromosome 1:151,982,915–154,870,281 (top) and zebrafish chromosome 16:22,682,280–29,387,215 (bottom) identified by ENSEMBL. Arrows denote relative gene length and orientation. Human *S100A8, S100A9,* and *S100A12* are shown in orange; zebrafish *s100s* and their human orthologs are shown in blue. The non-*S100* genes *adar/ADAR* and *KCNN3* are diagnostic for the syntenic region. Not all genes in the region are depicted. **(C)** A multiple sequence alignment of human S100A8, S100A9 and S100A10 compared to the zebrafish s100s from a similar genomic context. At the top, secondary structure features of human S100A8 are shown. Under this, the bar is colored by amino acid conservation from the alignment (blue=low, red=high). The consensus sequence from all sequences in the alignment is shown above the individual protein sequences. Amino acids found in a particular position in at least 50% of the sequences shown are shaded. The antigen for the "Fish Calprotectin" antibody was raised against the peptide boxed in yellow. **(D)** A pairwise percent identity matrix of S100 protein sequences from C (darker box indicates higher identity; S1 Table). S100 pair identity values for zebrafish s100s compared to human S100A8 range from 21.4–34.4%, where both s100a10b and s100t score 34.4%. Values comparing zebrafish proteins to S100A9 range from 23.8–39.6% with the highest identity shared with s100a1.

   

the zebrafish genomic region most similar to human chromosome Chr 1:152-155M, which encodes 19 of the 24 human *S100s*, including *S100A8* and *S100A9*. This region corresponded to zebrafish chromosome 16 (S1 Fig). Specifically, human Chr 1:154.6M-154.7M and zebrafish Chr 16:23.5M-23.7M cover the *KCNN3* and *ADAR* genes adjacent to tandem repeats of *S100* genes in both species (Fig 1B).

The existence of this shared cluster indicates that a handful of *S100* genes were in this genomic context at least in the bony vertebrate ancestor ~430 million years ago, as established in previous work [33]. The syntenic relationships, however, also give evidence for extensive evolution after the divergence of bony fishes and tetrapods: the orientation and placement of genes are different, and several orthologs to human *S100s* are missing from this genomic location but present on other zebrafish chromosomes. Further, most of the zebrafish *s100s* in this region appear to be teleost-specific duplicates [33]. This includes *ictacalcin (icn), icn2, s100t, s100s,* and *s100w*. The only clear orthologs to human genes are zebrafish s100a10b, co-orthologous to both human *S100A10* and *S100A11*, *s100v1*, co-orthologous to both human *S100A13* and *S100A14*, and *s100a1*, orthologous to human *S100A1* (Fig 1A and B).

## Human calprotectin and zebrafish s100 protein sequences share low sequence identity

Previous work identified zebrafish s100a10b as calprotectin using human S100A8 as a query in a BLASTP search against the zebrafish proteome [30]. In line with their results, we found that using human S100A8 as a BLASTP query against the NCBI RefSeq database (v. 224) yielded s100a10b as the top zebrafish hit (e-value: 8E-17, percent identity: 36.4%). The zebrafish icn2 and icn proteins were the next most similar hits. We assessed the quality of the hit by reciprocal BLAST, meaning we used the zebrafish s100a10b protein sequence as a query against the human proteome on NCBI. This yielded human S100A1 (9E-34; 55.9%) as the top hit, not S100A8. In fact, S100A8 (5E-16; 36.4%) and S100A9 (4E-18; 42.5%) were far down the hit list, following human proteins S100A10, S100Z, S100P, S100A11, S100B, S100A4, S100A12, S100A5, S100A6, and S100A2. This is consistent with previous phylogenetic analyses that place S100A8 and S100A9 as distant paralogs to zebrafish s100a10b (Fig 1A).

To evaluate sequence similarity and identity, we aligned zebrafish s100 protein sequences from the syntenic region to human S100A8, S100A9, and S100A10 sequences (Fig 1C). As expected, there is high conservation at sites that form the EF-hand and pseudo-EF-hand calcium-binding domains of the S100 proteins. There is low conservation in the region connecting the two EF-hands and at the termini. We next determined the sequence identity shared between zebrafish s100s and human S100A8, S100A9, and S100A10 (Fig 1D; S1 Table). We find similarly low levels of shared identity between human S100A8 and S100A9 and the zebrafish s100s. Overall, there is no obvious candidate zebrafish s100 that is like calprotectin by sequence similarity or identity.

The sequence alignment also allowed us to ask what zebrafish s100 protein(s) might be recognized by the commercially available "Fish Calprotectin" ELISA Kit from MyBioSource. This kit was made with antibodies raised against a 20 amino acid partial peptide of a human calprotectin-like protein (GenBank: AAB33355.1), which forms the N-terminal helix and beginning of the EF-hand 1 domain of human S100A8 (Fig 1C, yellow box). Eight to nine residues in this helix are highly conserved in several zebrafish s100s including s100b, s100a10b, s100a10a, s100a1, s100z, s100s and s100w as well as several other unrelated proteins. This suggests the antibody may have broad, non-specific interactions with multiple zebrafish proteins.

## Could a calprotectin paralog have convergently evolved similar immune functions?

We hypothesized that, although there is not a direct calprotectin ortholog in zebrafish, there could be a paralogous s100 that convergently evolved immune functions like calprotectin in zebrafish and other teleost fish. Like the analogous evolution of wings in bats and birds, the emergence of similar immune functions in two different *s100* genes would support that divergent species experienced similar environmental pressures and evolved parallel but distinct solutions to this pressure.

 

Zebrafish share six *s100* orthologs with mammals as well as eight *s100* genes unique to teleost fish [33] (Fig 1A). In the following sections, we will address the possibility that a s100 protein distantly related to calprotectin could have convergently evolved a calprotectin-like role in the zebrafish immune response.

**Single cell RNA sequencing dataset analyses identify zebrafish *s100s* upregulated in immune cells during an immune response**

Our bioinformatic analyses revealed zebrafish do not have a calprotectin ortholog; however, it is possible that a different s100 protein convergently evolved calprotectin-like activity. To investigate this possibility, we identified zebrafish s100s that share a similar expression profile to calprotectin. Calprotectin is expressed constitutively in mammalian neutrophils, monocytes, and several epithelial cell types and is upregulated upon infection and injury [40,41]. We queried existing zebrafish single cell RNA sequencing (scRNAseq) datasets for zebrafish *s100* genes expressed in immune cells and upregulated in response to injury.

We used the UCSC cell browser to visualize a zebrafish development dataset (NCBI Bioproject: 564810) [42,43] and assessed immune cell expression of *s100* genes in whole fish 1-, 2-, and 5-days post-fertilization (Table 1; S2 Fig). Of the genes that share genomic context with *S100A8* and *S100A9* (Fig 1B), we found that *s100a10b, icn, icn2,* and *s100a1* are expressed in immune cells of developing zebrafish. *S100w, s100t,* and *s100v1*, also in this genomic region, show very low expression in immune cell clusters. Four *s100* genes from other zebrafish chromosome locations show some expression in immune cells: *s100v2, s100u, s100z,* and *s100a11*. Finally, the remaining annotated zebrafish *s100* genes—*s100s, s100b,* and *s100a10a*—appear in very few immune cells at the 1-, 2-, and 5-day stage of development.

To evaluate whether expression of these genes are upregulated during the innate immune response to injury, we explored a scRNAseq dataset tracking local responses after tail fin amputation (NCBI GEO accession number GSE137971) [44]. Cells were isolated from several 1-year-old (adult) zebrafish caudal fins at the time of amputation (pre-injury) and at 1-, 2-, and 4-days post-amputation (dpa) from the same position.

In this dataset, the macrophage marker *mpeg1.1* is expressed above background in a subset of each cell cluster described in the dataset (epithelial, mesenchymal, and hematopoietic) and is highly expressed in the hematopoietic cluster. In response to injury, *mpeg1.1* expression slightly decreases at 1dpa, but then surpasses pre-injury levels in days 2 and 4 (S3 Fig). In contrast, neutrophil marker *mpx* is not detected above background in the hematopoietic cluster; it is only upregulated in a small number of basal epithelial and mesenchymal cells of regenerating fins 4dpa (S3 Fig). *Lyz*, another neutrophil marker, is found to be highly expressed in a single hematopoietic cell pre-injury but is not detected

**Table 1. Single-cell RNAseq profiles for zebrafish *s100s* syntenic to human calprotectin.**

| Gene name | Linkage Group and Synteny | Ortholog Call | Developmental dataset | Injury dataset |
|---|---|---|---|---|
| *s100a1* | Chr 16 ~syntenic | Z, A1 | Immune cells | Expressed in some hematopoietic cells, increased post-injury |
| *s100a10b* | Chr 16 syntenic | A10 | Immune cells | Highly expressed in hematopoietic cells, post-injury expression similar to mpeg1.1 |
| *s100w* | Chr 16 syntenic | A13, A14, A16 | Low in immune cells | Expressed in some hematopoietic cells, post-injury expression similar to mpeg1.1 |
| *icn* | Chr 16 syntenic | A1 | Immune cells | Highly expressed in hematopoietic cells, post-injury expression similar to mpeg1.1 |
| *icn2* | Chr 16 syntenic | A1 | Immune cells | Expressed in several hematopoietic cells, post-injury expression similar to mpeg1.1 |
| *s100t* | Chr 16 ~syntenic | A1 | Low in immune cells | **Not enriched** in hematopoietic cells, expression slightly increased post-injury |
| *s100v1* | Chr 16 ~syntenic | A13, A14 | Low in immune cells | **Not enriched** in hematopoietic cells, expression unchanged post-injury |

above baseline post-amputation. These data suggest injury by fin clip does not induce a strong transcriptional response by neutrophils at the site of amputation.

With the exception of *s100t* and *s100v1*, zebrafish *s100* genes that share genomic context with *S100A8* and *S100A9* were enriched in hematopoietic cell populations, with *s100a1* to a lesser degree. *S100a10b* and *s100a1* exhibit immediate and persistent upregulation after injury, whereas *icn, icn2,* and *s100w* initially drop in expression and then return roughly to pre-injury levels by 4dpa, similar to *mpeg1.1* (Table 1; S3 Fig). Zebrafish *s100* genes from other chromosomal loci show varying levels of expression in hematopoietic cells and in the injury response (S3 Fig). Combining the synteny and scRNAseq analyses, we determined that zebrafish *s100a10b, s100a1, s100w, and icn* were promising candidates to begin functionally comparing to calprotectin, though this is only a subset of immune-related zebrafish *s100* genes.

### Recombinant zebrafish s100 proteins fold and interact with calcium

We chose to functionally characterize the zebrafish s100 proteins which seemed most promising to behave like calprotectin based on genomic context and gene expression profiles: s100a10b, s100a1, s100w and icn. We left out icn2 because it is very similar to icn by all metrics including sequence identity (87.37%: they differ by 14 amino acids, 8 of these at the termini; Fig 1). The protein sequences and plasmids used for recombinant expression are shown in S2 Table.

We started by structurally characterizing the four selected zebrafish proteins. We used AlphaFold2 to predict structures for all four proteins [45–47]. Overlaying the predicted structures with the crystal structure of human calprotectin (RCSB ID 5W1F) shows high predicted structural similarity (Fig 2A). High α-helical content is a shared feature of all known S100 proteins, as well as the predicted zebrafish s100 structures. We tested whether this held for our selected zebrafish s100s recombinantly expressed and purified from *Escherichia coli* (*E. coli*). We measured their secondary structure content by far-UV circular dichroism (CD). This revealed signal minima at 208 and 222 nm consistent with primarily α helical structures (Fig 2B).

Most S100 proteins bind calcium and undergo a conformational change exposing a hydrophobic binding surface [48,49]. We tested whether this held for the four zebrafish s100 proteins by measuring calcium-induced changes in protein secondary and tertiary structure by far-UV CD and intrinsic fluorescence, respectively. We found that all four recombinantly expressed zebrafish proteins exhibited evidence of calcium-induced conformational change (Fig 2B and C).

Upon addition of saturating calcium (orange), zebrafish s100a10b and s100a1 exhibited an increase in helical content, while s100w and icn, in contrast, show little change in secondary structure (Fig 2B). The intrinsic fluorescence of all four proteins, however, responded to calcium (Fig 2C). Intrinsic fluorescence captures changes in the local chemical environments of tyrosine and tryptophan residues, suggesting that calcium binding induces a change in the tertiary structure of all measured s100 proteins. This is consistent with the canonical calcium-induced rotation of the third helix relative to the other helices of S100s [48].

Taken together, these results show that these four zebrafish s100 proteins are folded, bind to calcium, and undergo the calcium-induced conformational changes expected for members of the family.

### Zebrafish s100s do not exhibit nutritional immunity characteristics like human calprotectin

One of the most important biological functions of human calprotectin is antimicrobial activity via nutritional immunity: calprotectin binds to and sequesters transition metals, thus inhibiting bacterial growth. We evaluated the antimicrobial abilities of each of the four zebrafish s100 proteins against human-derived *Stapholococcus epidermidis* and zebrafish-derived *Vibrio ZWU0020* and *Aeromonas ZOR001* strains. *S. epidermidis* was previously shown to be susceptible to human calprotectin [10,50]; the response of the zebrafish-derived strains is unknown, however, both strains elicit immune responses in zebrafish [51]. Fig 3A shows the dose-dependent antimicrobial activity of human calprotectin against each strain over 13 hours in nutrient rich media across three biological replicates. For all three strains, increasing amounts of calprotectin (from blue to green) leads to decreased final $OD_{600}$ values (indicated by black arrows), confirming that the previously uncharacterized zebrafish commensal bacteria are susceptible to calprotectin's metal sequestration. A positive control for the effects of metal chelation on each strain by EDTA is included in Supplemental S4 Fig.

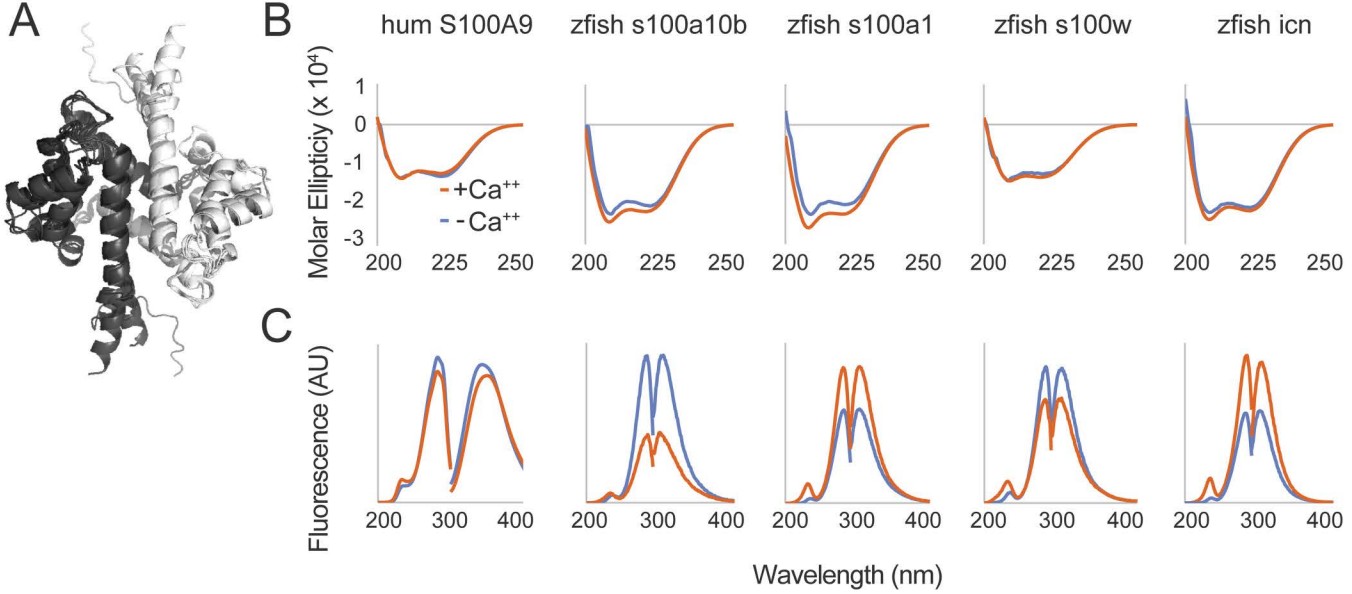

**Fig 2. Recombinantly expressed zebrafish s100 proteins are folded and respond to calcium. (A)** Overlaid AlphaFold2 structure predictions for all zebrafish s100 homodimers in this dataset, as well as the human S100A8/S100A9 heterodimer (5W1F RCSB ID). Different chains of each homodimer are shown in black or white. **(B)** Far UV circular dichroism spectra for each protein in the presence of 2 mM Ca++ (orange) and then adding 5mM EDTA (blue). Units are in molar ellipticity (deg × cm²/dmol) over wavelength (nm). **(C)** Fluorescence excitation and emission spectra for each protein in the presence or absence of calcium (orange and blue, respectively). The fluorescence units are arbitrary; the x-axis is wavelength in nanometers. Excitation spectra were collected while observing fluorescence at the maximum emission wavelength; emission spectra were collected while exciting at the maximum excitation wavelength.

Each bacterial strain exhibited unique growth phases and phase-dependent responses to calprotectin (Fig 3A). We wanted to evaluate the overall nutritional immunity effect of calprotectin across all phases of growth for each strain, rather than comparing $OD_{600}$ at snapshots in time. To do this, we first calculated the area under the curve (AUC) for all $OD_{600}$ curves from 0–13 hrs. A stronger antimicrobial effect results in a smaller AUC. We quantified the consequence of increasing the calprotectin challenge on overall bacterial growth by calculating the difference in the AUC value for growths at each concentration of calprotectin against the unchallenged control growth (ΔAUC). We then generated dose-response curves for each strain by plotting ΔAUC as a function of protein concentration. A negative ΔAUC value indicates growth inhibition, while a zero or positive value indicates no antimicrobial activity. The results showed that increasing the calprotectin dose increases overall growth inhibition of all three bacterial strains (Fig 3B, yellow curves). This reproduces what the literature already knows about the S. epidermidis response to calprotectin and reveals that zebrafish-derived *Aeromonas ZOR001* and *Vibrio ZWU0020* are susceptible to nutritional immunity by calprotectin's metal chelation properties.

To assess the nutritional immunity capacity of the zebrafish s100s, we performed identical experiments and analyses. The $OD_{600}$ dose response curves for zebrafish s100s are show in S4 Fig. We calculated ΔAUC curves for each bacterial strain under increasing concentrations of each s100 (Fig 3B). Unlike the effect of human calprotectin, none of the four zebrafish s100 proteins exhibited nutritional immunity (Fig 3B; S4 Fig). Human calprotectin was the only protein to exhibit nutritional immunity under these conditions (Fig 3B, yellow curve). Bacteria treated with zebrafish s100a1 showed improved growth relative to bacteria in the absence of s100 (Fig 3B, brown curve). Zebrafish s100a10b increased growth of *S. epidermidis* and had no effect on growth of zebrafish-derived bacterial strains (Fig 3B, orange). Zebrafish s100w (purple) improved *S. epidermidis* growth, showed a possible slight inhibitory effect on *Aeromonas ZOR001* for concentrations at or above 50 µM, and did not affect *Vibrio ZWU0020* growth at any concentration. Similarly, zebrafish icn (blue)

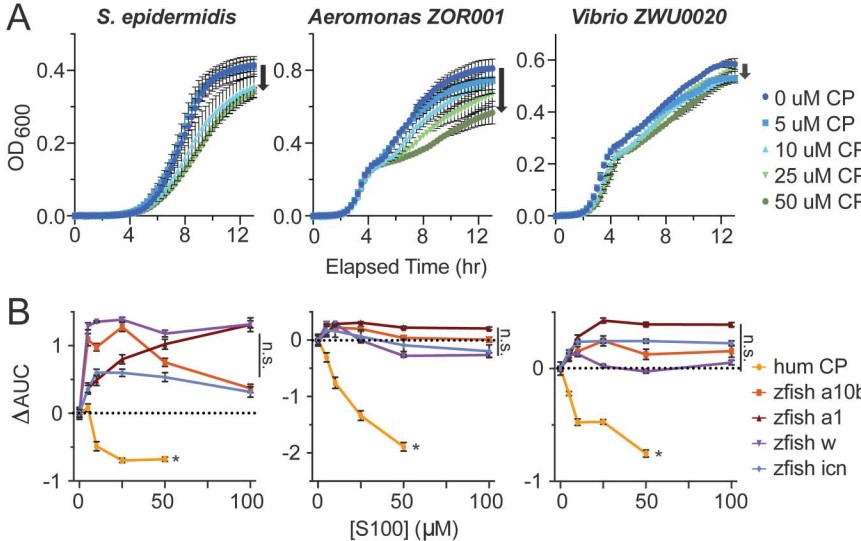

**Fig 3. Zebrafish s100s do not exhibit nutritional immunity activity like human calprotectin. (A)** Dose dependence of human calprotectin challenge on human and zebrafish commensal bacteria. Each column specifies which bacteria was used for the set of nutritional immunity assays: human-derived Gram-positive *Stapholococcus epidermidis* and two zebrafish-derived Gram-negative bacteria, *Aeromonas* strain**:** *ZOR001* and *Vibrio* strain: *ZWU0020*. Bacterial growth was measured by $OD_{600}$ over 13 hours with challenge by increasing doses of human calprotectin noted in the legend on the right. Concentration increases from dark blue to dark green and black arrows indicate how growth is affected as calprotectin increases. Error bars indicate standard error of three biological replicates. **(B)** Zebrafish s100 dose effects on human and zebrafish commensal bacterial growth compared to human calprotectin. Each datapoint shows the change in the area under the curve from the absence of s100 protein to the indicated s100 concentration, measured from growth curves like those shown in panel A (S4 Fig). The dotted line at zero represents no effect on bacterial growth.

improved *S. epidermidis* growth. We performed an ANOVA analysis on the antimicrobial activity at 50 µM protein for every condition and found that human calprotectin had a statistically different effect than any of the zebrafish s100 proteins ($p < 0.0039$), but that the zebrafish proteins could not be distinguished from one another (S3 Table).

### Zebrafish s100s do not exhibit pro-inflammatory activity like S100A9

Antimicrobial activity is not the only function of mammalian calprotectin. Extracellular calprotectin is thought to act as a cytokine, activating immune receptors like the Receptor for Advanced Glycation Endproducts (RAGE) and Toll-like receptor 4 (TLR4) [20]. However, RAGE evolved in mammals from a family of cell adhesion molecules [52], so is not biologically relevant to zebrafish s100 function. TLR4 is an ancient receptor that arose in the ancestor of all bony vertebrates [53,54]. The S100A9 component of calprotectin has been demonstrated to drive TLR4 activation, inducing nuclear localization of NF-κB and transcription of a wide variety of pro-inflammatory proteins [55,56]. This activity can be reproduced in an *in vitro* functional assay by transfecting human embryonic kidney-293T (HEK293T) cells with plasmids encoding the proteins of the TLR4 complex (TLR4, MD-2, and CD14), as well as a plasmid placing luciferase behind an NF-κB promoter. We treat the cells with exogenous S100A9 and measure the expression and activity of luciferase produced in response [37].

We wanted to see if our purified zebrafish s100 proteins could play a similar pro-inflammatory role; therefore, we tested the ability of these proteins to activate TLR4 in this assay (Fig 4). Zebrafish have three ohnologs of tetrapod *TLR4*: *tlr4ba, tlr4bb,* and *tlr4al* [53,54]. Zebrafish Tlr4ba has been shown to induce inflammation in response to endotoxin, the small molecule lipopolysaccharide (LPS) derived from Gram-negative bacterial outer membranes, but neither Tlr4bb nor Tlr4al showed activity [53].

We validated our assay by testing the ability of each complex to activate in response to endotoxin, the canonical agonist for the receptor (blue, outlined). As expected, the human TLR4 and zebrafish Tlr4ba complexes responded strongly to endotoxin. As observed previously [53], Tlr4bb and Tlr4al did not show signal above vehicle treatment (light blue). We next

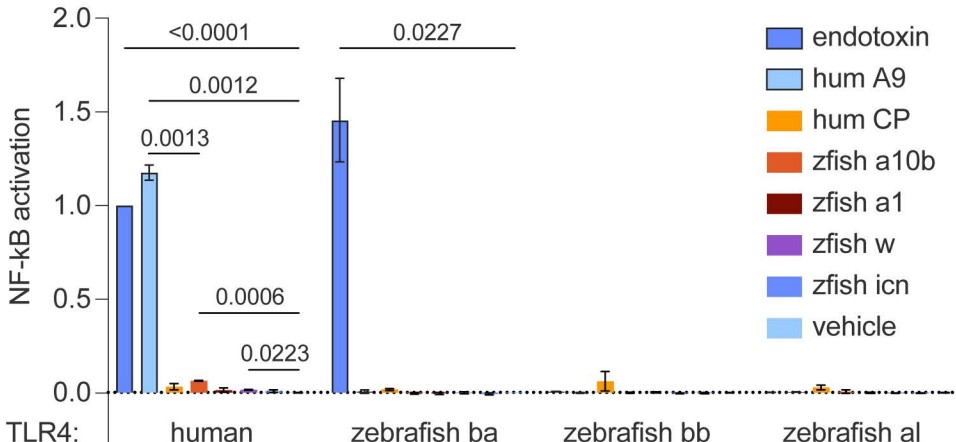

**Fig 4. Zebrafish s100s do not exhibit the pro-inflammatory characteristics of S100A9.** The bar graph shows relative activation of human and zebrafish TLR4 complexes in the presence of 1 µM zebrafish s100 proteins. The positive controls for this experiment included human TLR4 and zebrafish Tlr4ba treated with endotoxin (blue, outlined), and human TLR4 treated with 1 µM human S100A9 (light blue, outlined). There is no known agonist for zebrafish Tlr4bb and Tlr4al complexes. The negative control was vehicle treatment with 200 ng/uL Polymyxin B (PB) in PBS. All data was background subtracted and normalized to the signal from human TLR4 treated with endotoxin. Bars show the average signal across three biological replicates, with error bars indicating standard error. Statistics shown are p-values <0.05 from unpaired Welch t-tests.

challenged all four complexes with human S100A9 (light blue, outlined) and calprotectin (yellow). We show that although pure calprotectin is not pro-inflammatory in our assay, 1 µM human S100A9 potently activates human TLR4, as expected, but none of the zebrafish Tlr4 complexes. This suggests that sterile inflammation via a mechanism similar to S100A9 activation of TLR4 is not a conserved function in zebrafish.

We next tested the ability of zebrafish s100s to activate each TLR4 complex at 1 µM. We observed no statistically significant agonist activity for any protein against any zebrafish Tlr4. We detected a small amount of signal for treatment with zebrafish s100a10b (orange) and zebrafish s100w (purple) against human TLR4; however, this was narrowly above background and much lower than the response for either endotoxin or human S100A9. We repeated the same experiment with 10 µM zebrafish s100s and observed no convincing agonist activity (S5 Fig).

These zebrafish s100 proteins show no evidence of activity against human TLR4 nor the three zebrafish Tlr4 ohnologs. Given the potent response of human TLR4 and zebrafish Tlr4ba to positive controls (human S100A9 and/or endotoxin), and that false positives are common in this assay due to endotoxin contamination from recombinant protein expression in *E. coli*, the lack of signal thus gives strong evidence that these zebrafish s100s cannot activate TLR4 in the same fashion as human S100A9. This does not rule out the possibility that they might activate a yet uncharacterized zebrafish immune receptor, but if such a receptor cannot be activated by calprotectin, S100A8, or S100A9 then this activity would also not be characterized as convergent evolution.

### Structure predictions do not support that convergent evolution of a calprotectin-like metal chelation site occurred in zebrafish s100s

We next wanted to extend our analysis beyond the four zebrafish s100 homodimers we experimentally characterized. There are ten additional zebrafish *s100* genes in the genome. These may form homodimers, heterodimers, or even high-order complexes that play functional roles in fish physiology and innate immunity. Experimentally measuring the functional properties of all 196 possible zebrafish s100 homodimers and heterodimers would, however, be a massive undertaking. We therefore took a computational approach.

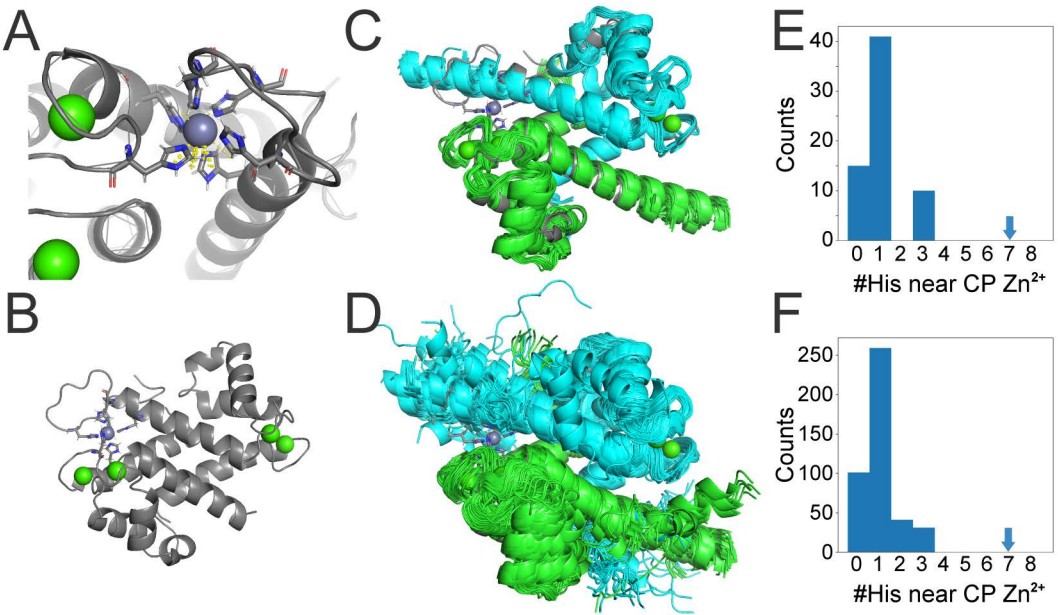

**Fig 5. Predicted structures of zebrafish s100 homodimers and heterodimers.** **(A)** A close-up of calprotectin's (gray) hexa-histidine site coordinating a zinc ion (purple sphere) via polar contacts (dashed yellow lines). **(B)** A view of the calprotectin heterodimer from crystal structure PDB: 8JSC. Calcium ions are represented as green spheres. **(C and D)** are overlay views of AlphaFold3 model structure predictions of homodimers and heterodimers, respectively, aligned to the calprotectin heterodimer. The first chain of each model dimer is in blue, and the second chain is in green. This excludes the s100t homodimer (S7 Fig). **(E and F)** Histograms showing the number of predicted homodimer **(E)** and heterodimer **(F)** models that have a certain number of histidine alpha carbons within 9 Å of the calprotectin $Zn^{2+}$ ion. Arrows indicate that human calprotectin has seven histidines within 9 Å of the $Zn^{2+}$ ion.

We chose to structurally assess all zebrafish s100 homo- and hetero-dimers for possible convergently evolved antimicrobial activity by looking for calprotectin-like hexa-histidine sites. Because the structural basis by which S100A9 activates TLR4 remains unknown, we only assessed predicted antimicrobial function.

The crystal structure of human calprotectin bound to $Zn^{2+}$ (PDB: 8SJC) illustrates that the metal-chelating hexa-histidine site is constructed from two residues from human S100A8 (H17 and H27) and four residues from human S100A9 (H87, H91, H99 and H101) (Fig 5A).

For our computational analysis, we used AlphaFold3 [57] to make five model predictions for each zebrafish homo- and hetero-dimer in the presence of four calcium ions and one zinc ion. We used the full-length sequence for all zebrafish proteins, except for s100u, for which we used only residues 1–89. (Positions 90–240 of s100u extend beyond the calprotectin dimer alignment, are predicted to be disordered, and have no histidine residues that could chelate metals as part of a hexa-histidine site.) After generating our structural models for all 196 possible zebrafish s100 homo- and heterodimers, we aligned them to the crystal structure of $Zn^{2+}$ bound calprotectin dimer (PDB: 8SJC; chains A and C). All predicted dimers, except for the s100t homodimer, aligned well with calprotectin. (In the presence of zinc, s100t is predicted to form a unique S100 dimer fold which may be interesting to explore in the future (S7 Fig)).

We next quantified the number of histidine residues per predicted model within 9 Å of the calprotectin-bound $Zn^{2+}$ ion (Fig 5). This is a generous cutoff that would allow for conformational rearrangements to bring histidine residues into proximity with the ion binding site. For reference, the human calprotectin structure has seven histidine residues within 9 Å of its bound $Zn^{2+}$ ion.

Across our five models for each of the 196 homo- and heterodimers, we observed no more than three histidine residues within 9 Å of the calprotectin $Zn^{2+}$ ion (Fig 5E-F). 87% of the structural models had zero or one histidine within this

cutoff; the remaining 13% had two or three histidine residues. All the proteins we functionally characterized in the nutritional immunity assay—s100a1, s100a10b, s100w, and icn—are predicted to have a single histidine near the calprotectin hexa-histidine site. This is consistent with their lack of observed antimicrobial activity.

Our computational analysis suggests that none of the zebrafish homo- or hetero-dimer s100 complexes convergently evolved calprotectin's hallmark high-affinity metal chelating hexa-histidine site. While this finding does not exclude antimicrobial activity via some other mechanism, this would be entirely convergent, and thus, a separate evolutionary function than what is seen in mammalian calprotectin.

## Discussion

Employing zebrafish as a model for studies of innate immunity and host-microbe interactions is a promising field of work. As vertebrates, zebrafish share much of their physiology and immune defense mechanisms with humans [58], thus enabling mechanistic insight into health, disease, and the host-microbe interface. However, when approaching this research, we must be cognizant of the more than 400 million years since our most recent common ancestor, allowing for species-specific differences evolved to cope with diverse environments and other pressures.

We assert here that recent studies and commercial products have made the incorrect assumption that calprotectin exists in the fish innate immune response. We re-evaluate phylogenetic evidence to look for an ortholog of calprotectin in fish and confirm that zebrafish do not share an s100 protein within the clade containing mammalian calprotectin and therefore do not have a vertically inherited ortholog of calprotectin.

We also probed the hypothesis that a fish s100 protein from a similar genomic context to calprotectin might have convergently evolved calprotectin-like innate immune functions. We characterized the nutritional immunity and pro-inflammatory activity of four zebrafish s100s, s100a1, s100w, icn, and the previously studied zebrafish "calprotectin" s100a10b, in assays which are normally used to test calprotectin function. None of the zebrafish proteins performed like human calprotectin. We note that there may be alternate conditions under which these zebrafish s100s could elicit antimicrobial or pro-inflammatory effects; perhaps by growing bacteria in minimal media or at a higher calcium concentration for the nutritional immunity assay, or testing pro-inflammatory abilities with different zebrafish immune receptors. But, given the evolutionary histories of these genes (i.e., zebrafish do not have an ortholog of calprotectin), such functions—if they exist—must have evolved convergently.

These experimental results on a limited number of zebrafish s100 proteins are likely general across the genome, at least at the level of antimicrobial activity. This is because our computational models of all possible zebrafish s100 homo- and heterodimers displayed no evidence of a hexa-histidine metal chelation site that would exert nutritional immunity effects like calprotectin.

Although s100a10b is not calprotectin, researchers found that transcription of s100a10b increases in response to infection. This increase in expression is also reflected in the zebrafish injury scRNAseq dataset. But rather than viewing this as a convergent calprotectin or S100A8 activity, it seems best to view this through the lens of the mammalian S100A10 inflammatory response. This is because s100a10b is orthologous to S100A10; any shared activity between the human and zebrafish proteins likely reflects shared ancestry. Like zebrafish s100a10b, mammalian S100A10 is upregulated during inflammatory responses. While the function of zebrafish s100a10b is unknown, in mammals S100A10 regulates plasminogen-dependent macrophage migration via interactions with annexin A2. Based on our analyses, we believe such a role is a much more likely role for s100a10b than a convergently evolved calprotectin activity.

This illustrates a broader point that future research on zebrafish s100s and other proteins could benefit from anchoring their hypotheses in orthology. We cannot prove a negative: our work does not demonstrate that no zebrafish s100 protein exists that supplies some subset of the functions of human calprotectin. But if a zebrafish s100 has antimicrobial or pro-inflammatory activity, it must have evolved that activity convergently and independently from mammalian calprotectin. Further, and importantly, such a protein does not shed direct light on mammalian biology. A convergent zebrafish

s100 calprotectin-like protein would help us understand zebrafish biology and would be intriguing from the perspective of protein evolution. But it would almost certainly have a different regulatory scheme and subset of the human calprotectin functions.

A limitation of this study is that we did not test all possible zebrafish s100s because we focused on homodimer proteins that seemed most promising to share properties of convergent evolution with calprotectin. Zebrafish s100 proteins represent a largely uncharacterized set of proteins, many of which evolved in the ray-finned fishes [33]. Studies of other heterodimeric human S100 complexes have been done and prove to have altered functions [59]. In the future, this type of analysis could be done with zebrafish proteins to explore whether a heterodimer state confers nutritional immunity or pro-inflammatory activity. However, there is currently no evidence to suggest this is likely. We are excited to see how investigations of s100 innate immune roles continue with the development of new inflammation models in zebrafish, including chemically induced colon inflammation models and pathogenic bacterial and viral infection models [60,61]. We remain intrigued by the idea that convergent evolution may exist between mammalian calprotectin and some other protein(s) in zebrafish.

We conclude that it is crucial that we use an evolutionary lens and careful biochemical analyses to probe homology between zebrafish and human proteins so that we can make accurate extrapolations of findings from zebrafish models of human biology.

## Materials and methods

### Protein sequence analysis

In our BLAST and reciprocal BLAST study, we used the NCBI web interface and BLASTP with default parameters to query the NCBI RefSeq database (v. 244). We aligned and calculated S100 sequence identity using Clustal Omega [62]. We viewed and generated figures displaying the alignment and consensus sequence in SnapGene Viewer 5.2.4.

### Single-cell RNA sequencing datasets

We used interactive data visualization platforms for existing zebrafish single cell RNA sequencing (scRNAseq) datasets to look for zebrafish *s100* genes expressed in immune cells and upregulated in response to injury (Table 1). We used the UCSC cell browser to visualize a zebrafish development dataset (NCBI Bioproject: 564810) [42,43] which sampled whole fish 1-, 2-, and 5-days post-fertilization (S2 Fig). We then used a web platform built by the authors, Hou et al., (https://k326xh.shinyapps.io/FinRegenerationSCRNA/) to access their fin amputation and regeneration dataset (NCBI GEO accession number GSE137971) [44] (S3 Fig). Cells were isolated from the same position in the caudal fin of several 1-year old (adult) zebrafish at the time of amputation (pre-injury) and at 1-, 2-, and 4-days post-amputation (dpa). Differential gene expression for both datasets was determined by comparing the gene expression profile of each cell with those of the rest of the cells in the dataset using a Wilcoxon rank sum test in Seurat.

### Protein purification

We purchased all zebrafish *s100* genes from GenScript in the pET-28a(+)-TEV vector with an N-terminal 6x-Histidine tag and TEV protease cleavage site (S2 Table). All genes were codon-optimized for expression in *E. coli*. We expressed human *calprotectin* (with *S100A8* containing the C42S mutation *and S100A9* containing the C3S mutation) and human *S100A9*/C3S in a pET-Duet vector without purification tags. We transformed Rosetta2(DE3)pLysS *E. coli* cells with plasmids. We used transformant glycerol stocks to inoculate cultures in 15 mL Luria broth (LB) with 50 µg/mL kanamycin and 34 µg/mL chloramphenicol. We incubated cultures overnight at 37 °C, shaking at 250 rpm. The following day, we diluted 15 mL saturated cultures into 1.5 L of LB with antibiotics. When the $OD_{600}$ reached 0.6–1.0, we induced recombinant protein expression with 1 mM IPTG and 0.2% glucose and then grew overnight at 16 °C, shaking at 250 rpm. We pelleted

cells at 3,000 rpm for at least 15 minutes in an F6B rotor in a Beckman Coulter preparative centrifuge. We stored pellets at -20 °C for up to one month.

We prepared protein lysates for purification with the following method: We vortexed pellets (6–9 g) in 45 mL buffer from the first chromatography step (see below) until cells were resuspended, added 15 µL each of DNase I and Lysozyme (ThermoFisher Scientific), and incubated at room temperature with gentle shaking for at least 10 minutes. We lysed the cells by sonication at 55% amplitude with 0.3 second pulse on, 0.7 second pulse off, for 3–5 minutes. We pelleted cell debris by centrifugation at 15,000 rpm at 4 °C for at least 20 minutes in a JA-20 rotor in a Beckman Coulter preparative centrifuge and collected the supernatant. To remove remaining large debris, we filtered lysate supernatant through a 0.2 µm pore syringe filter immediately prior to purification chromatography.

We purified all proteins using an Äkta PrimePlus Fast Protein Liquid Chromatography system using two stacked 5 mL HiTrap columns at each step. We used HisTrap FF columns for Ni-affinity and Q HP columns for anion exchange (GE Health Science). All chromatography was performed at 4 °C. At the end of purification, we confirmed protein purity was > 95% by SDS-PAGE (S6 Fig). Then, we dialyzed each protein overnight into 4 L of 25 mM Tris, 100 mM NaCl, pH 7.4 at 4 °C. We placed 2 g/L Chelex 100 resin (Bio-Rad) in the dialysis buffer to chelate divalent metal ions. We concentrated each protein to roughly 2 mg/mL and syringe-filtered through a 0.22 µm filter directly into liquid nitrogen to sterilize and flash freeze before storing at -80 °C.

We purified TEV-cleavable 6xHis-tagged zebrafish s100 proteins with the following scheme. We used 25 mM Tris, 100 mM NaCl, pH 7.4 buffer as the base for all chromatography buffers. We ran our protein lysate over a Ni-affinity column with a 50 mL wash and eluted over a 75 mL gradient from 25–1000 mM imidazole to collect proteins with strong Ni binding capacity. We determined which fractions contained our desired protein by SDS-PAGE and pooled these fractions. To separate our recombinant proteins from their Ni-binding His-tag, we added 5 mM DTT and 6xHis-tagged TEV protease to the pooled fractions and incubated the reaction at room temperature with gentle shaking for at least 5 hours. We then dialyzed the protein solution overnight into 4 L buffer with 25 mM imidazole and 5 mM DTT to allow cleavage to come to completion and to remove excess imidazole from the sample. We performed a second round of Ni-affinity chromatography. Without the His-tag, the zebrafish s100s have low affinity for Ni. Therefore, we isolated pure, non-tagged zebrafish s100 proteins at this step during a 50 mL wash in 25 mM imidazole and then used a step gradient to 1 M imidazole to elute His-tagged and other contaminant proteins that had higher affinity for the Ni column. Purified zebrafish s100s were prepared for storage as described above.

We purified human calprotectin using Ni-affinity chromatography at pH 7.4 and anion exchange at pH 8. When expressing calprotectin, S100A8 and S100A9 homodimers are also expressed and must be removed during chromatography. In the presence of calcium, S100A9 and calprotectin bind divalent metal ions like Ni, but S100A8 and most other lysate proteins do not. We loaded our Ni-affinity column with calprotectin lysate, washed away most A8 and contaminants in a 50 mL wash, and then eluted calprotectin and S100A9 over a 75 mL gradient from 0–1000 mM imidazole and 1–0 mM $CaCl_2$ in 25 mM Tris, 100 mM NaCl, pH 7.4 buffer. We pooled elution peak fractions containing calprotectin and contaminant S100A9 homodimers, as determined by SDS-PAGE, and dialyzed overnight in 4 L of 25 mM Tris, 100 mM NaCl at pH 8. We loaded our sample onto an anion exchange chromatography column in 25 mM Tris, 100 mM NaCl at pH 8 with 100 mM NaCl. Because S100A9 has a lower pI than calprotectin, it binds the anion column more strongly at pH 8. We used a 50 mL wash in 100 mM NaCl to isolate calprotectin and then used a step gradient increasing the salt to 1 M NaCl to remove S100A9 and other contaminants from the column. At this point, calprotectin was pure and prepared for storage as described, and fractions with S100A9 and other contaminants were discarded.

We used a similar protocol to purify human S100A9. We performed Ni-affinity chromatography as described for calprotectin. We then performed anion exchange chromatography using a 50 mL wash and collected fractions over a 70 mL gradient elution from 100–1000 mM NaCl in 25 mM Tris, pH 8 buffer to isolate S100A9 from contaminant proteins that also bind the anion column. We used SDS-PAGE to confirm fractions with S100A9, pooled and dialyzed these fractions

overnight into 4 L of 25 mM Tris, 100 mM NaCl at pH 6. As a final step, we loaded the S100A9 sample onto an anion exchange column in 25 mM Tris, 100 mM NaCl buffer at pH 6. S100A9 binds weakly to the anion column at pH 6. Therefore, we collected S100A9 in a 50 mL wash in 100 mM NaCl, and then removed contaminants from the column with a step elution at 1 M NaCl. Pure S100A9 was prepared for storage as described.

## Far-UV circular dichroism and fluorescence spectroscopy

Prior to biophysical measurements, we thawed and equilibrated all proteins into 25 mM Tris, 100 mM NaCl, pH 7.4 via overnight dialysis in 4 L buffer at 4 °C. We determined protein concentrations by Bradford Assay using bovine serum albumin (BSA) standards and the molecular weight of each dimeric structure, then diluted to ~ 10 µM in dialysis buffer.

For all spectroscopic measurements, we assessed metal-induced changes to the spectra by measuring the spectrum in the presence of 2 mM $CaCl_2$ and then adding excess EDTA at 5 mM and re-measuring the spectrum. We collected far-UV circular dichroism data between 200–250 nm using a J-815 CD spectrometer (Jasco) with a 1 mm quartz spectrophotometer cell (Starna Cells, Inc. Catalog No. 1-Q-1). We collected 3 scans for each condition, and then averaged the spectra and subtracted a blank buffer spectrum using the Jasco spectra analysis software suite. We converted raw ellipticity into mean molar ellipticity using the concentration and number of residues in each protein.

We collected intrinsic tyrosine and/or tryptophan fluorescence using the J-815 CD spectrometer (Jasco) with an attached model FDT-455 fluorescence detector (Jasco) using a 1 cm quartz cuvette (Starna Cells, Inc.). We collected a single excitation and emission scan at 10 nm/min with a 10 nm bandwidth, 1 nm data pitch, and 1 sec D.I.T. for each condition and then subtracted a blank buffer spectrum using the Jasco spectra analysis software suite. Depending on the sample signal, we set the detector sensitivity to either 630 or 800 Volts. We conducted excitation scans by measuring 305 nm light emitted at for all zebrafish proteins and 345 nm emitted light for human S100A9 for each excitation wavelength from 200–295 nm. For emission scans we used 280 nm light to excite zebrafish proteins and 288 nm for human S100A9, and measured light emitted at all wavelengths from 285–425 nm.

## Nutritional immunity assay

We measured the antimicrobial activity of zebrafish s100s and human calprotectin against human- and zebrafish-derived bacterial strains using a modified version of a well-established assay that will be described here [5,50,63,64]. Bacterial strains used in this assay include 1) *Staphylococcus epidermidis,* a human commensal strain previously shown to respond to calprotectin [50,64]; 2) *Aeromonas ZOR001*, isolated from zebrafish and not previously characterized for response to calprotectin; and 3) *Vibrio ZWU0020*, isolated from zebrafish and not previously characterized for response to calprotectin but related to human-derived *Vibrio cholerae* shown to respond to calprotectin [29]. We obtained both zebrafish-derived strains from the Guillemin lab at the University of Oregon.

Each week, we plated bacterial strains from glycerol stocks onto antibiotic-free LB agar and grew at 30 °C overnight before storing plates at 4 °C. The day before an experiment, we inoculated a 5 mL culture in liquid LB media with a single colony from each strain and grew overnight at 30 °C with shaking. The following day, we diluted cultures 1:100 in 5 mL LB and grew to an $OD_{600}$ around 0.8 by the time of the experiment. *Aeromonas ZOR001* and *Vibrio ZWU0020* were diluted 2 hours before the experiment. *S. epidermidis* grew more slowly so required dilution 4 hours prior to the experiment.

The day before each experiment, we thawed a single S100 protein from -80 °C, concentrated to at least 200 µM using a Nanosep 3K Omega spin concentrator (Pall Corporation), and dialyzed overnight at 4 °C into 4 L of Experimental Buffer (25 mM Tris, 100 mM NaCl, pH 7.4) with 2 g/L Chelex 100 resin (Bio-Rad) to chelate residual transition metal ions. After dialysis, we filter-sterilized the protein through a Ultrafree-MC-VV centrifugal filter with Durapore PVDF 0.1 µm and kept at 4 °C until time of experiment.

To start the experiment, we made a protein dilution series by mixing a desired amount of protein in sterile Experimental Buffer with the appropriate amount of LB to achieve a ratio of 62:38, respectively. We then brought the volume of these protein solutions up to 1.7 mL in Experimental Media (EM). We made EM by mixing Experimental Buffer in LB at a ratio of 62:38, respectively, and filter-sterilized. We distributed each sample in aliquots of 160 µL across ten wells of a clear Falcon 96-Well, Cell Culture-Treated, Flat-Bottom Microplate. At this time, we diluted each bacterial strain to an estimated $OD_{600}$ of.008 in 5 mL Experimental Media with calcium (EMC). We made EMC by adding 10.2 µM $CaCl_2$ to EM, and sterile-filtered. Then, we added 40 µL of dilute bacteria or EMC without bacteria (contamination control) to each well, bringing the final volume to 200 µL/well, and making technical triplicate conditions for bacterial strains. To counteract sample evaporation, the outermost wells of the plate contained 160 µL EM and 40 µL EMC, and we wrapped the plate in a single layer of parafilm.

We measured bacterial growth by $OD_{600}$ every 15 minutes over 13 hours in a Molecular Devices SpectraMax i3. The plate was shaken for 5 seconds before the first read, then for 10 minutes between each subsequent read. We set the plate reader temperature to 25 °C, however, over the course of the overnight growth, the actual temperature reached 37 °C. The final concentration of metals in the media without bacteria was measured using ICP-MS at the USR Elemental Analysis Core. The measured concentrations were Ni: 45.4 µM, Ca: 107.3 µM, Cu: 157.4 µM, Mg: 160.5 µM, Mn: 216.6 µM, Fe: 1.1 mM, and Zn: 5.9 mM.

For the analysis, we background subtracted each experimental condition using $OD_{600}$ values for the matching concentration of S100 protein concentration in buffer without bacteria added. We used Prism to average the replicates by condition, determine the standard error of the mean, and graph the results. We performed an ANOVA analysis on the antimicrobial activity at 50 µM protein for every condition and reported the results in Supplemental S3 Table.

## Pro-inflammatory activity assay

We tested the S100A9-like pro-inflammatory activity of zebrafish s100s using a well-established assay [37,50,53]. This assay measures relative activation of the TLR4-mediated immune response through NF-κB. For each experiment, we thawed all zebrafish s100 proteins and human S100A9 from -80 °C, buffer exchanged into endotoxin-free PBS, then treated with endotoxin removal spin columns (ThermoFisher Scientific) to remove residual LPS from the purification process.

We performed each experiment in technical triplicate and followed the Dual-Glo Luciferase Assay System protocol (Promega). We transiently transfected adherent HEK293T cells (ATCC; CRL-11268) in a Falcon 96-Well, Cell Culture-Treated, Flat-Bottom Microplate with pcDNA vector plasmids using PLUS and Lipofectamine Reagents (ThermoFisher Scientific). Plasmids contained genes for human or zebrafish TLR4 complex components and *Renilla* luciferase enzyme under constitutively active promoters, and the firefly luciferase gene controlled by an NF-κB promoter. For human TLR4 complex transfections, we transfected 10 ng human *TLR4*, 0.5 ng human *LY96* (*MD-2*), and 1 ng human *CD14* plasmids per well. For zebrafish Tlr4 complex transfections, we used 10 ng zebrafish *tlr4ba*, *bb*, or *al*, 20 ng zebrafish *ly96*, and 1 ng mouse *Cd14* plasmids per well, as this ratio gives us the best signal to noise ratio. Zebrafish do not have an annotated *CD14*. Previous studies have shown zebrafish Tlr4ba can be activated in the presence of mouse and human CD14, but more strongly with mouse. We also transfected all wells with 1 ng *Renilla* plasmid, 20 ng *elam-Luc* (firefly), and brought the total DNA mass per well to 100 ng with empty pcDNA vector in a total media volume of 200 µL per well.

After 20–24 hours incubation at 37 °C in 5% $CO_2$, we removed all 200 µL of transfection mix from each well. We then treated transfected HEK293T cells with 100 µL of one of the following treatment mixes: 1) 1 µM S100 protein and 200 ng/µL Polymyxin B to bind up LPS in media, 2) 0.1 ng/µL LPS-R (tlrl-eklps; Invivogen) as a positive control for human TLR4, 3) 1 ng/µL lipid IVa as a positive control for zebrafish Tlr4ba activation, or 4) PBS and 200 ng/µL Polymyxin B as a negative vehicle control treatment. Because there is no known activator of zebrafish Tlr4bb and Tlr4al complexes [53], we treated these transfected cells with 1 ng/µL lipid IVa for consistency. After incubating again at 37 °C in 5% $CO_2$ for 3–4 hours, we removed and discarded 60 µL of treatment mix from each well. We chemically lysed the cells by adding 30 µL

Dual Glo lysis reagent containing firefly luciferin and incubated in the dark for 7 minutes. We then mechanically lysed the cells by scraping the bottom of each well with a pipet tip and transferring 60 μL of cell solution to an opaque 96-well plate. After a 7-minute incubation in the dark at room temperature, we measured luminescence per well produced by firefly luciferase activity using a Molecular Devices SpectraMax i3. Then we added 30 μL of Dual-Glo Stop & Glo buffer containing firefly luciferase quencher and *Renilla* luciferase reagent, incubated for 7 more minutes, and measured luminescence.

For the analysis, we took the firefly signal for each experimental condition and background subtracted the averaged firefly signal of wells transfected with the corresponding complex but treated with buffer without agonist. We did the same for the *Renilla* signal, with background signal considered as the averaged signal from wells with same treatment condition but transfected only with vector. We divided the background-subtracted firefly signal for each well by the background-subtracted *Renilla* signal for that same well. To simplify comparisons across experiments, we normalized the firefly/*Renilla* value for each well to the triplicate average of the firefly/*Renilla* values for human TLR4 complex treated with 0.2 ng/μL LPS-R. Statistical analyses were done using the multiple unpaired Welch t-tests in Prism v10.3.1 and p-values were reported on the graph.

### Protein structure prediction and analysis

We collected zebrafish s100 polypeptide sequences identified in The Zebrafish Information Network (ZFIN). We fed these sequences into the AlphaFold3 web server [57] to build all possible homodimer and heterodimer complexes in the presence of four calcium ions and one zinc ion. We then opened the five output models from each AlphaFold3 prediction in Open-Source PyMol and aligned them to the crystal structure of calprotectin bound to zinc (PDB: 8JSC, chains A and C). We used PyMol to determine which histidine residues were within 9 Å of the calprotectin-bound zinc ion for each model. This information was used to make histogram plots for homodimer and heterodimer datasets. We also qualitatively noted when plausible alternate zinc-binding sites were predicted.

### Supporting information

**S1 Table. Identity matrix for S100 proteins used in this study.** Human S100 proteins (A8, A9, and A10) and homologs from zebrafish (a10b, w, icn, icn2, a1, t, and v1).
(TIF)

**S2 Table. Sequences of *s100* genes expressed and purified in this study.** Lowercase amino acids indicate the addition of a 6xHis tag and TEV protease cleavage site. Bold and underlined amino acids are mutations relative to the reference sequence.
(TIF)

**S3 Table. Results of an ANOVA analysis of zebrafish s100 protein antimicrobial activity.** The top sub-table shows the correlation of bacterial species (*S. epi.*, *A. ZOR001*, and *V. ZWU0020*) and protein (hCP, s100a1, s100a10b, ictacalcin, s100w) with the mean change in area under the growth curves for the 50 μM treatment condition. The bottom sub-table shows the results of a post hoc Tukey test applied to the ANOVA results. This test reveals that the effect of hCP is significantly different than any of the zebrafish proteins (p values between 0.00169 and 0.0039; bolded rows). The effects of the zebrafish proteins cannot be distinguished.
(TIF)

**S1 Fig. EMSEMBL identification of synteny between region of human S100 genes and the zebrafish genome.** Central chromosome is human chromosome 1; outer chromosomes are zebrafish chromosomes with regions syntenic to regions of human Chr1. The red box on Chr1 indicates the region containing 19 of the 24 human S100 genes (1:154584825–154708290). This is syntenic to zebrafish Chr16.
(TIF)

**S2 Fig. Single-cell transcription of zebrafish *s100* genes in developing immune cells.** Developmental scRNA-seq datasets taken from the UCSC cell browser NCBI Bioproject: 564810. **A)** This set of 3 atlases shows the layout that will be used for panel B. Each point is a cell from 1-, 2-, or 5-days post-fertilization zebrafish, separated by transcriptional profile along the UMAP1 and UMAP2 axes. The "Full Atlas" on the far left shows how all the cells in the dataset are related to one another; the middle- and right-most atlases zoom in on the regions corresponding to macrophage clusters 71 and 184 and neutrophil cluster 150. These regions are shown by the small boxes on the full atlas. **B)** Each row of atlases corresponds to a different zebrafish *s100* RNA (labeled on the left). Cells are colored by the relative expression level of that RNA shown by the legend at the top (blue: no expression; red: high expression). **C)** A dot plot summarizes which zebrafish *s100s* are expressed in the macrophage and neutrophil clusters. Darkness of dot color represents average expression. Dot size indicates the percent of cells within the cluster that express the RNA. The authors performed differential gene expression analysis using the FindAllMarkers function in Seurat v3.4.4 using Wilcoxon rank sum test.
(TIF)

**S3 Fig. Single-cell transcription of zebrafish s100 genes in response to fin clip injury.** Data are taken from NCBI GEO accession number GSE137971 [44]. We accessed the data published at https://k326xh.shinyapps.io/FinRegenerationSCRNA/ on June 11th, 2024. The authors state that relative RNA expression level was found using Seurat v3.0 by comparing the expression profiles of the specified gene with those of the rest of the cells using Wilcoxon rank sum–based approach with the criteria of log fold change more than 0.25 and a minimum cell percentage of 0.25. **A)** The relative expression level of specific RNAs (y-axis) within certain cell types (x-axis; also denoted by color in legend) across all stages, pre-injury through 4dpa. Individual cells are shown as points. **B)** The relative expression of each s100 RNA (y-axis) within a specific stage of regeneration, e.g., pre-injury, 1-day post-amputation (dpa), 2dpa, or 4dpa (x-axis; also denoted by color in legend) across all cell types. **C)** Dot plots summarize the data in **B**. Dot size represents the percent of cells within the cluster expressing the gene. Dot color indicates the average expression level of the particular RNA within the cluster compared to its expression level in cells in the rest of the dataset using a Wilcoxon rank sum–based approach.
(TIF)

**S4 Fig. Nutritional immunity assays of zebrafish s100s.** Rows show assays done with the bacterial strain indicated on the left. Columns show results for the S100 protein indicated at the top. Bacterial growth was measured over 13 hours in the presence of S100 concentrations ranging from 0–100 μM, dark blue to dark green as shown in the legend at the top left. A control experiment was done using EDTA at concentrations equal to the ratio of the number of metal binding sites per calprotectin heterodimer. 6 binding sites total per dimer: 4 calcium-binding sites and 2 zinc-binding sites. All measurements were done in biological triplicate of technical triplicates except s100A1 at 100 μM which only contains data from two biological replicates. Datapoints and error bars represent the mean and standard error of biological replicates.
(TIF)

**S5 Fig. Pro-inflammatory assay with zebrafish s100 concentrations increased 10-fold.** Bars show the average signal across three biological replicates, with error bars indicating standard error. The positive controls for this experiment included human TLR4 and zebrafish Tlr4ba treated with endotoxin (green), and human TLR4 treated with 1 μM human S100A9 (yellow). For zebrafish experiments, we used 10 μM protein. There is no known agonist for zebrafish Tlr4bb and Tlr4al complexes. All data was background subtracted and normalized to the signal from human TLR4 treated with endotoxin.
(TIF)

**S6 Fig. Protein purity analysis.** A) The chromatogram of the final anion exchange chromatography step for the purification of human calprotectin. Calprotectin flowed through the column during the low salt wash step while S100A9 and other

proteins stuck to the column and were eluted later in high salt buffer. B) An SDS-PAGE analysis of fractions corresponding to the chromatogram (red numbers). Fractions 8–27 (underlined) show a ~ 1:1 ratio of S100A8 (bottom band) to S100A9 (top band) and were pooled for use as pure calprotectin. C) A gel showing purified human S100A9 and zebrafish s100a1, s100a10b, s100w, and icn. The ladder in both gels is the Spectra™ Multicolor Broad Range Protein Ladder. S100 proteins fall between ~10 kDa (green ladder band at bottom) and ~15 kDa (lowest blue ladder band).
(TIF)

**S7 Fig. Structural analysis of zebrafish s100t.** A) The calprotectin heterodimer (gray) from crystal structure PDB: 8JSC. Calcium ions are represented as green spheres and the zinc ion as a purple sphere. B) Zebrafish s100t homodimer aligned to calprotectin heterodimer. C) Zebrafish s100t homodimer. D) A close-up of calprotectin's hexahistidine site coordinating a zinc ion via polar contacts (dashed yellow lines). E) A close-up of the zebrafish s100t homodimer's potential alternate metal binding site. Sidechains predicted to form the interaction with a zinc ion are shown as sticks with polar contacts shown.
(TIF)

**S1 File. Raw data for circular dichroism and fluorescence spectroscopy.**
(XLSX)

**S2 File. Raw data for nutritional immunity assays.**
(XLSX)

**S3 File. Raw data for pro-inflammatory activity assays.**
(XLSX)

## Acknowledgments

We would like the thank the Guillemin lab at the University of Oregon for the zebrafish-derived bacterial strains used in this work, as well as the META Center for Systems Biology group for their guidance on experiments and interpretation of data.

## Author contributions

**Conceptualization:** Kona N. Orlandi, Michael J. Harms.

**Data curation:** Kona N. Orlandi.

**Formal analysis:** Kona N. Orlandi, Michael J. Harms.

**Funding acquisition:** Kona N. Orlandi, Michael J. Harms.

**Investigation:** Kona N. Orlandi.

**Methodology:** Kona N. Orlandi, Michael J. Harms.

**Project administration:** Kona N. Orlandi, Michael J. Harms.

**Resources:** Michael J. Harms.

**Supervision:** Michael J. Harms.

**Visualization:** Kona N. Orlandi.

**Writing – original draft:** Kona N. Orlandi.

**Writing – review & editing:** Kona N. Orlandi, Michael J. Harms.

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
