## [Decision Letter · Decision Letter 0]

2 Aug 2024

PONE-D-24-26499Zebrafish do not have calprotectinPLOS ONE

Dear Dr. Orlandi,

Thank you for submitting your manuscript to PLOS ONE. After careful consideration, we feel that it has merit but does not fully meet PLOS ONE’s publication criteria as it currently stands. Therefore, we invite you to submit a revised version of the manuscript that addresses the points raised during the review process. Please submit your revised manuscript by Sep 16 2024 11:59PM. If you will need more time than this to complete your revisions, please reply to this message or contact the journal office at plosone@plos.org . Please include the following items when submitting your revised manuscript:

We look forward to receiving your revised manuscript.

Kind regards,

Andrei Chernov

Academic Editor

PLOS ONE

Journal Requirements:

Reviewers' comments:

Reviewer's Responses to Questions

**Comments to the Author**

1. Is the manuscript technically sound, and do the data support the conclusions?

Reviewer #1: Yes

Reviewer #2: Partly

2. Has the statistical analysis been performed appropriately and rigorously? 

Reviewer #1: Yes

Reviewer #2: N/A

3. Have the authors made all data underlying the findings in their manuscript fully available?

Reviewer #1: Yes

Reviewer #2: Yes

4. Is the manuscript presented in an intelligible fashion and written in standard English?

Reviewer #1: Yes

Reviewer #2: Yes

5. Review Comments to the Author

Reviewer #1: The Zebrafish (Danio rerio) is widely used as model of human disease including inflammatory diseases to study immune responses and efficacy of drugs, owing to a well conserved innate immune system between zebrafish and humans, and the discovery of mediators and receptors that mediate inflammatory responses. The mammalian EF hand superfamily of S100 proteins with homeostatic intracellular functions, are known to mediate innate immune responses when expressed extracellularly. They act as damage associated molecular pattern (DAMP) molecules that bind to cell surface receptors to promote inflammatory signaling responses. Well-known among these are the S100A8 and S100A9 heterodimeric proteins, known as calprotectin, which serve both as mediators and biomarkers of inflammation and cancers. Multiple S100 proteins are expressed in zebrafish, and although there is no ortholog of S100A8/A9 proteins in the zebrafish, a few recent studies identify zebrafish s100a10b as calprotectin equivalent, given sequence similarity and upregulation during inflammation. A commercial antibody that could have non-specific reactivity against multiple zebrafish S100 proteins is marketed as fish calprotectin antibody.

In this manuscript the authors seek phylogenetic, biological and functional evidence to identify a calprotectin-like activity amongst zebrafish S100 proteins and conclude that zebrafish do not express calprotectin. Their review of phylogenetic studies and comparative synteny analysis reveal no calprotectin ortholog in zebrafish; assessment of zebrafish developmental scRNAseq data set and acute fin injury regeneration scRNAseq dataset available online show no functional equivalent of calprotectin. They choose zebrafish s100a10b, s100a1, s100w and icn, which show the highest gene expression profile similar to calprotectin, to recombinantly express and purify from E.coli and show that the proteins fold and bind to calcium, but do not exhibit the nutritional immunity activity of calprotectin, and do not promote NF-kappaB signaling in cells expressing human TLR4 or three zebrafish tlr4 ohnologs. Their work questions the assumption made by recent studies and commercial antibodies that a calprotectin-like activity could mediate some of the innate immune responses in zebrafish. The authors address an important issue given that zebrafish models of inflammation are increasingly being used to study innate immune responses. However, there are a few caveats that need to be addressed as indicated below, at least in text revision, even if not experimentally,

Comments:

The authors make a definitive statement in the title and in the text that zebrafish do not express calprotectin, without providing more extensive evaluation, even as they place the onus on other investigators in the field to prove that such a function exists. They evaluate the functions of only four of the nearly 14 or more S100 genes in zebrafish, some of which could be teleost-specific without ortholog equivalents in mammals. Without further extensive studies it cannot therefore be concluded that they mediate or do not mediate innate immune responses akin to calprotectin. At minimum, the title and discussion must be modified to be more reflective of the limitations of the current study as suggested below.

A. Physical injury models of acute inflammation in zebrafish have limitations, and therefore other models of inflammation including chemically induced colon inflammation models, and pathogenic bacterial and viral infection models are being developed and validated in zebrafish which could provide more in-depth analysis of the role of zebrafish s100 proteins in promoting innate immune responses.

B. It has recently been shown that zebrafish express a low-sensitivity Tlr4/Md-2 complex that confers immune response to LPS, (although there is no ortholog in zebrafish of CD14, a co-receptor required for TLR4 signaling). Based on this report, and since calprotectin is known to bind to TLR4, the authors evaluate NF kappa B signaling in response to purified human S100A9 or purified zebrafish s100 proteins on cells expressing human TLR4 or three zebrafish tlr4 ohnologs, and show that the zebrafish s100 proteins do not elicit tlr4 mediated intracellular signaling. However, there are other cell surface receptors such as RAGE, CD36, and GPCRs that bind to S100A8/S100A9 and promote intracellular signaling. There is some evidence that orthologs of these receptors, showing conserved synteny, do exist in zebrafish.

1. Minor comments:

a. How were the purities of the expressed zebrafish s100 proteins assessed?

b. Fig 1c: the peptide antigen sequence used to raise the fish calprotectin antibody is indicated to be boxed in yellow, but the boxed sequence could not be identified in the figure.

Reviewer #2: In this manuscript, Orlandi and Harms investigate whether an ortholog of mammalian calprotectin exists in zebrafish or not, based on phylogenetic analysis of the zebrafish S100 genes and evaluation of the antimicrobial and inflammatory properties of putative calprotectin gene candidates.

Zebrafish S100A10b was indeed previously reported to act as a possible surrogate of mammalian calprotectin, although not being a strict ortholog of the S100A8/A9 heterocomplex. To evaluate if this is truly the case, the authors of this manuscript first performed thorough phylogenetic and synteny analyses of S100 genes in vertebrates and demonstrated that the mammalian A8/A9/A12 clade was generated from gene duplication of a common ancestor gene in amniotes and evolved separately from other S100 groups in other vertebrates, including from the A10/A11 clade to which bony fish S100A10b belongs. Having confirmed that no strict orthology exists between calprotectin and any of the zebrafish S100 proteins, they further tested whether functional orthology may occur, focusing on the S100 proteins expressed in immune cells that they identified on the basis on published scRNA seq data. They first performed antimicrobial assays on three different bacterial pathogens (1 human and 2 zebrafish-specific), taking calprotectin as a reference, and showed that none of the investigated zebrafish S100 could elicit bactericidal activity. Next, they employed a cell-based assay to evaluate the TLR4-dependent activation of NFκB, and again showed that in contrast to human S100A9 and endotoxins, none of the zebrafish S100 tested could induce NFκB activation.

This nice piece of work further corroborates that a bona fide calprotectin protein does not exist in zebrafish. This is indeed an important question to answer as calprotectin is a valuable biomarker and target for anti-inflammatory applications in humans, and its investigation in alternative animal models like zebrafish could have held interesting promises from both an experimental and therapeutic point of view if the protein had been found in this organism. Furthermore, previous studies have misleadingly attributed the name calprotectin to S100A10b, suggesting a clear parallelism in function between these two proteins although it does not seem to be the case. I therefore find that this manuscript makes a very neat and important point in demonstrating that this assumption is wrong.

The manuscript is well written and pleasant to read. Most of the conclusions drawn are well supported by the data. I however believe that some additional experiments and controls are necessary to fully ascertain the absence of a functional ortholog of calprotectin in zebrafish, as several important points have not been explored by the authors and thus leave the question still open in my opinion. I therefore recommend that the points listed below are further investigated before this manuscript can be considered for publication in PLoS ONE:

1) As the authors rightly pointed out in the discussion, calprotectin is an heterodimer and it has been unambiguously shown in mammals that the A8 and A9 homodimers have distinct properties than the heterocomplex, both for inflammatory and antimicrobial properties. I therefore believe that a proper comparison cannot bypass this point and heterodimerization of zebrafish S100 proteins, at least for the ones present in immune cells and investigated in this study, must be evaluated. This can be easily monitored using the split GFP trap method as described in Spratt et al FEBS J 2018 (doi:10.1111/febs.14775). If any of these proteins are able to form heterodimers, then the antimicrobial and NFκB-activation assays should be repeated with the heterocomplexes.

2) For the antimicrobial assay, the authors chose three representative pathogens. While Vibrio species have already been investigated in zebrafish for their link with S100 proteins, the two others haven’t. It is in my opinion important to also test E. tarda for which a clear modulation of S100 genes in response to infection has already been observed.

3) I find the representation of the data for the antimicrobial assays not very intuitive, especially the choice of measuring the area under the growth curve and displaying the ΔAUC value as a measure of growth inhibition by S100 proteins is a bit odd. Why not just choose to represent the 600OD after a given incubation period against increasing concentrations of S100? This might be easier to interpret.

4) The antimicrobial function of mammalian S100 proteins, including calprotectin, is primarily believed to occur through metal sequestration. In the assay, the maximal concentration of S100 used was 100 µM. But the content in divalent cations of the media, as measured by ICP-MS and reported in the Methods section, is much higher than that. Zinc and iron, two central nutrients for pathogens, are for example present at 5.9 and 1.1 mM concentrations, respectively. This means that the S100 metal binding sites would be saturated, at least partly. While a weak antimicrobial activity is detected for calprotectin, such concentrations in divalent cations may hinder the true bactericidal activity of zebrafish S100 proteins if they have lower affinity for metal ions than calprotectin. The authors should provide some controls here:

4a) First, a chelator of divalent cations like EDTA/EGTA should be used at the same concentration as the S100 proteins for positive control of bacterial growth inhibition.

4b) Second, the assay should be reproduced with optimized growth media or with higher concentrations of S100 proteins to detect any potentially weak antimicrobial activity.

5) The antimicrobial activity of calprotectin is linked to its unique hexahistidine site that chelates metal ions with high affinity. The authors chose to focus on zebrafish S100 proteins that are expressed in immune cells. But are there any zebrafish S100 proteins that may resemble more calprotectin for this particular function, and notably possess a high content of histidine residues in their C-terminal region? This should be investigated at least computationally and discussed in the manuscript.

6) In the cell-based assay for NFκB activation, the authors use human S100A9 as the positive control representing mammalian S100s. Since the entire study is based on calprotectin, I find it odd that calprotectin is not employed in this assay as well. The experiment should be repeated with calprotectin for proper comparison.

7) TLR4 is not the sole receptor for S100A8/A9 in mammals. Obviously, RAGE is not present in zebrafish. But what about the other receptors, especially those that may preferentially bind the S100A8 moiety of calprotectin? Their presence or not in zebrafish should be brought to discussion.

8) No statistical analyses are provided in the manuscript. This is an important flaw of the study, in particular for the antimicrobial assays. It is important to confirm that the decrease in 600OD observed in the presence of CP for the 3 tested pathogens is statistically significant. Such statistical analyses should therefore be provided with clear description of the statistical tests employed.

9) Many papers describe the purification of calprotectin from co-expression system as a very straightforward procedure. However, they rarely demonstrate that they obtain an heterodimer preparation devoid of contamination by A8 or A9 homodimers. The chromatogram for the purification of calprotectin on anion exchange chromatography should be displayed in Supplementary material, together with the SDS-PAGE analysis of the fractions corresponding to the A9 homodimer and the proposed A8/A9 heterodimer, for assessment of the purity and homogeneity of the protein samples.

Minor comment:

10) Clear description of what is the ‘vehicle’ in Figure 4 should be mentioned either in the legend of the figure or in the Method section describing the NFκB activation assay.

6. PLOS authors have the option to publish the peer review history of their article (what does this mean? ). If published, this will include your full peer review and any attached files.

**Do you want your identity to be public for this peer review?** For information about this choice, including consent withdrawal, please see our Privacy Policy .

Reviewer #1: **Yes: ** Dr. Geetha Srikrishna

Reviewer #2: **Yes: ** Laure Yatime

---

## [Author Response · Author response to Decision Letter 0]

7 Dec 2024

Reviewer 1: Comment: The authors make a definitive statement in the title and in the text that zebrafish do not express calprotectin, without providing more extensive evaluation, even as they place the onus on other investigators in the field to prove that such a function exists. They evaluate the functions of only four of the nearly 14 or more S100 genes in zebrafish, some of which could be teleost-specific without ortholog equivalents in mammals. Without further extensive studies it cannot therefore be concluded that they mediate or do not mediate innate immune responses akin to calprotectin. At minimum, the title and discussion must be modified to be more reflective of the limitations of the current study as suggested below. (--suggestions moved later in responses: noted as Comment 1A and 1B)

Response: Thank you for pointing this out. We have updated the title, several lines of text, and added a section (line 212, subheading: “Could a calprotectin paralog have convergently evolved similar immune functions?”). These changes reflect that we know for certain that there is no ortholog of calprotectin in zebrafish, but we cannot rule out that a s100 or other protein may have convergently evolved functions akin to calprotectin. We have emphasized the nuance between proving a protein is the same protein in different species, versus establishing convergent functional properties between different proteins. We hope we have now made this important distinction clear and that we drive home the point that zebrafish did not inherit an ortholog of the calprotectin protein and therefore do not have calprotectin.

We have also modified the text to highlight that the experiments in this manuscript test only the possibility of functionally convergent evolution in a fraction of the potential zebrafish s100 protein complexes. There is much work to be done on this front; the volume of work to rigorously establish the various roles of zebrafish s100 homo- and hetero-dimers and their implications in health and disease, let alone their antibacterial and proinflammatory abilities, could fill a series of manuscripts. We have now elaborated on this in the text in a new section (line 418, subheading “Structure predictions do not support convergent evolution of a calprotectin-like metal chelation site in zebrafish s100s”) as well as in the discussion (line 518).

Reviewer 2: Comment 1) As the authors rightly pointed out in the discussion, calprotectin is an heterodimer and it has been unambiguously shown in mammals that the A8 and A9 homodimers have distinct properties than the heterocomplex, both for inflammatory and antimicrobial properties. I therefore believe that a proper comparison cannot bypass this point and heterodimerization of zebrafish S100 proteins, at least for the ones present in immune cells and investigated in this study, must be evaluated. This can be easily monitored using the split GFP trap method as described in Spratt et al FEBS J 2018 (doi:10.1111/febs.14775). If any of these proteins are able to form heterodimers, then the antimicrobial and NFκB-activation assays should be repeated with the heterocomplexes.

Response: We have considered experiments to explore zebrafish heterodimer s100 activities. There are two reasons for which we chose not to pursue this:

1) Due to the lack of phylogenetic evidence that any zebrafish s100 is orthologous to S100A8 or S100A9, we focused on testing the functional relevance of previous claims that zebrafish s100a10b is calprotectin—s100a10b cannot be calprotectin due to its phylogenetic history but could have convergently evolved metal chelating and/or immune stimulatory abilities that could be relevant to disease model research in zebrafish.

2) As this reviewer pointed out in comment #9, in order to form a functional calprotectin complex, both S100A8 and S100A9 must be expressed from the same plasmid in E. coli; you cannot simply mix two preparations of homodimers. This means that even just testing all combinations of the four promising zebrafish s100s candidates we identified would require expressing, purifying, and functionally testing six additional protein complexes.

As noted by both reviewers, there are at least 14 s100s in zebrafish. Due to the complicated differences in the evolutionary history of mammalian and teleost s100s, it is not obvious which heterocomplex would be the most likely to have convergently evolved bactericidal and/or proinflammatory properties. In our previous response to reviewer 1 we noted that a large a body of work, which is out of the scope for this manuscript, would be to experimentally explore the functions of all 196 zebrafish s100 homo- and hetero-complexes.

We have now included a section describing a rough structural evaluation of potential zebrafish s100 dimer complexes using AlphaFold3 and PyMol (see line 418, subheading: “Structure predictions do not support convergent evolution of a calprotectin-like metal chelation site in zebrafish s100s”).

Our first pass efforts to reveal convergent evolution within the zebrafish s100 complexes was not fruitful but does not diminish the fact that there is no ortholog of calprotectin in zebrafish (i.e. zebrafish do not have “calprotectin”). We have attempted to drive this point home more clearly throughout the text.

Reviewer 2: Comment 5) The antimicrobial activity of calprotectin is linked to its unique hexahistidine site that chelates metal ions with high affinity. The authors chose to focus on zebrafish S100 proteins that are expressed in immune cells. But are there any zebrafish S100 proteins that may resemble more calprotectin for this particular function, and notably possess a high content of histidine residues in their C-terminal region? This should be investigated at least computationally and discussed in the manuscript.

Response: Thank you for this interesting suggestion. The crystal structure of human calprotectin bound to zinc (PDB: 8SJC) illustrates the residues that make up the hexa-histidine site that this reviewer refers to. The residues that constitute the hexa-histidine site are two residues from human S100A8: H17 and H27 within the N-terminal helix and loop region, as well as four residues from human S100A9: H87, H91, H99 and H101 at the C-terminus.

In an alignment of all teleost s100s from Kraemer et al., 2008 (Additional File 4), s100b (including Danio rerio s100b) has 3 histidine residues at its C-terminus which is the highest concentration out of all teleost s100 proteins. Zebrafish s100b also has two N-terminal histidine residues at positions 16 and 39. We have investigated this idea more thoroughly using the AlphaFold3 structure prediction software. We have included an evaluation of all predicted structures of homo- and hetero-dimers and highlighted potential sites for metal chelation in the new results section, “Structure predictions do not support convergent evolution of a calprotectin-like metal chelation site in zebrafish s100s” (line 418), included a new figure, “Fig 5. Predicted structures of zebrafish s100 homodimers and heterodimers” (line 434), and added supplementary figure S7.

Reviewer 1: Comment 1A. Physical injury models of acute inflammation in zebrafish have limitations, and therefore other models of inflammation including chemically induced colon inflammation models, and pathogenic bacterial and viral infection models are being developed and validated in zebrafish which could provide more in-depth analysis of the role of zebrafish s100 proteins in promoting innate immune responses.

Response: An insightful suggestion. We have now included references to reviews on zebrafish inflammation models [line 526].

Reviewer 2: Comment 2) For the antimicrobial assay, the authors chose three representative pathogens. While Vibrio species have already been investigated in zebrafish for their link with S100 proteins, the two others haven’t. It is in my opinion important to also test E. tarda for which a clear modulation of S100 genes in response to infection has already been observed.

Response: It would be interesting to explore whether zebrafish s100s previously noted to be transcriptionally upregulated during the immune response to infection by the serious fish pathogen, Edwardsiella tarda, have any ability to directly affect E. tarda growth. For example, Zhang et al., 2019 noted that transcripts of s100a1, s100a10b, icn1, icn2, s100b, and s100v1 were significantly upregulated in the gut and spleen 3-6 hours post-infection with E. tarda. To our knowledge there has not been a direct study on E. tarda’s response to calprotectin as it is not a major human pathogen and zebrafish do not have calprotectin, therefore it was not considered a standard to be tested against.

Although this could be useful knowledge for the field of fish pathogen research, we do not believe this particular experiment is necessary to prove that the four studied zebrafish s100 proteins lack the broad-spectrum antimicrobial activity of metal ion chelation which calprotectin employs. In our antimicrobial assay, we show that human calprotectin can inhibit growth, in a dose-dependent manner, of three bacterial strains. We used these strains in part because they were accessible at our institution. The human-derived S. epidermidis was chosen because it is a standard for testing calprotectin functional variants across many studies, and the two zebrafish-derived bacterial strains are related to human pathogens and elicit immune responses in zebrafish. We’ve updated the text to clarify this in line 312.

Reviewer 2: Comment 3) I find the representation of the data for the antimicrobial assays not very intuitive, especially the choice of measuring the area under the growth curve and displaying the ΔAUC value as a measure of growth inhibition by S100 proteins is a bit odd. Why not just choose to represent the 600OD after a given incubation period against increasing concentrations of S100? This might be easier to interpret.

We wanted the reader to be able to directly compare differential effects of each protein on overall bacterial growth rather than at a snapshot in time. The Vibrio panel of Figure 3A illustrates that the endpoint OD600 reading is not always an accurate representation of the inhibitory effect on all phases of bacterial growth.

Response: We thought about alternatively graphing the average growth rate for each strain challenged with increasing concentrations of all proteins, which might be easier to interpret, but would add significant noise to the data.

We also considered simply showing the growth curves of every strain/protein combination, as shown in Supplement Figure 4. However, this felt like it would be an overwhelming figure.

We have now included a more detailed explanation of the ΔAUC values and how to interpret them starting at line 334 in the results section “Zebrafish s100s do not exhibit nutritional immunity characteristics like human calprotectin”.

Reviewer 2: Comment 4) The antimicrobial function of mammalian S100 proteins, including calprotectin, is primarily believed to occur through metal sequestration. In the assay, the maximal concentration of S100 used was 100 µM. But the content in divalent cations of the media, as measured by ICP-MS and reported in the Methods section, is much higher than that. Zinc and iron, two central nutrients for pathogens, are for example present at 5.9 and 1.1 mM concentrations, respectively. This means that the S100 metal binding sites would be saturated, at least partly. While a weak antimicrobial activity is detected for calprotectin, such concentrations in divalent cations may hinder the true bactericidal activity of zebrafish S100 proteins if they have lower affinity for metal ions than calprotectin. The authors should provide some controls here:

4a) First, a chelator of divalent cations like EDTA/EGTA should be used at the same concentration as the S100 proteins for positive control of bacterial growth inhibition.

4b) Second, the assay should be reproduced with optimized growth media or with higher concentrations of S100 proteins to detect any potentially weak antimicrobial activity.

Response: This (4a) is a good experimental control. We initially used calprotectin as our positive control but have now included the suggested experiment in our supplemental data (S4 Fig) for completeness. This experiment was done with EDTA concentrations that match the concentration of metal binding sites in each calprotectin heterodimer challenge. The calprotectin heterodimer has 4 calcium-binding sites and 2 transition metal binding sites, so 6 uM EDTA to every 1 uM calprotectin. This control suggests that the calprotectin metal-binding sites were not saturated in our growth assays, likely due to non-saturating levels of calcium which increase calprotectin’s metal binding affinity. Nonetheless, our conclusion is the same that each strain is inhibited in growth by calprotectin under these conditions, but not by any of the zebrafish s100s tested.

These (4b) are good suggestions; however, we do not believe they are necessary. Our data shows that under the conditions in our assay, human calprotectin can inhibit growth of each bacterial strain in a dose-dependent manner.

Increasing the zebrafish s100 dose to 100 uM dimer greatly exceeds the most extreme physiological extracellular calprotectin level of ~40 uM (1 mg/mL) reported in humans (Johne et al., 1997). Farr et al., 2022 reported zebrafish “calprotectin” levels up to ~2.4 uM (60 ng/mL) after infection with Vibrio cholerae. Our results show that even 100 uM concentrations did not exhibit antimicrobial effects for any zebrafish s100. Most of the deltaAUC curves for these proteins plateau before reaching 100 uM, suggesting that additional protein would not change our conclusion.

As our new AlphaFold3 structure predictions confirm, it is unlikely that the zebrafish proteins have a strong metal chelating site which would be necessary to support the convergent evolution of calprotectin’s nutritional immunity function.

Reviewer 2: 6) In the cell-based assay for NFκB activation, the authors use human S100A9 as the positive control representing mammalian S100s. Since the entire study is based on calprotectin, I find it odd that calprotectin is not employed in this assay as well. The experiment should be repeated with calprotectin for proper comparison.

Response: This is a valid concern. In the HEK-293T cell-based assay, S100A9 is a potent activator of TLR4, and pure calprotectin is not. For this reason, we compared the TLR4 agonist ability of zebrafish s100 proteins against the well-characterized S100A9 agonist. We have now included a comparison to human calprotectin for completeness in (Figure 4, yellow bars).

Reviewer 1: Comment 1B. It has recently been shown that zebrafish express a low-sensitivity Tlr4/Md-2 complex that confers immune response to LPS, (although there is no ortholog in zebrafish of CD14, a co-receptor required for TLR4 signaling). Based on this report, and since calprotectin is known to bind to TLR4, the authors evaluate NF kappa B signaling in response to purified human S100A9 or purified zebrafish s100 proteins on cells expressing human TLR4 or three zebrafish tlr4 ohnologs, and show that the zebrafish s100 proteins do not elicit tlr4 mediated intracellular signaling. However, there are other cell surface receptors such as RAGE, CD36, and GPCRs that bind to S100A8/S100A9 and promote intracellular signaling. There is some evidence that orthologs of these receptors, showing conserved synteny, do exist in zebrafish.

+

Reviewer 2: Comment 7) TLR4 is not the sole receptor for S100A8/A9 in mammals. Obviously, RAGE is not present in zebrafish. But what about the other receptors, especially those that may preferentially bind the S100A8 moiety of calprotectin? Their presence or not in zebrafish should be brought to discussion.

Response: This is an excellent point made by both reviewers. We have attempted to get a human RAGE reporter system working in our lab and were not successful. We have now included discussions on the hypothesis that alternate zebrafish immune r

---

## [Decision Letter · Decision Letter 1]

14 Jan 2025

PONE-D-24-26499R1Zebrafish do not have a calprotectin orthologPLOS ONE

Dear Dr. Orlandi,

Thank you for submitting your manuscript to PLOS ONE. After careful consideration, we feel that it has merit but does not fully meet PLOS ONE’s publication criteria as it currently stands. Therefore, we invite you to submit a revised version of the manuscript that addresses the points raised during the review process. **One reviewer still has two minor concerns that need to be addressed before the revised manuscript can be accepted.**

We look forward to receiving your revised manuscript.

Kind regards,

Andrei Chernov

Academic Editor

PLOS ONE

**Journal Requirements:**

**Additional Editor Comments:**

One reviewer still has two minor concerns that need to be addressed in the revised manuscript.

Reviewers' comments:

Reviewer's Responses to Questions

**Comments to the Author**

1. If the authors have adequately addressed your comments raised in a previous round of review and you feel that this manuscript is now acceptable for publication, you may indicate that here to bypass the “Comments to the Author” section, enter your conflict of interest statement in the “Confidential to Editor” section, and submit your "Accept" recommendation.

Reviewer #1: All comments have been addressed

Reviewer #2: (No Response)

2. Is the manuscript technically sound, and do the data support the conclusions?

Reviewer #1: Yes

Reviewer #2: Yes

3. Has the statistical analysis been performed appropriately and rigorously? 

Reviewer #1: Yes

Reviewer #2: Yes

4. Have the authors made all data underlying the findings in their manuscript fully available?

Reviewer #1: Yes

Reviewer #2: Yes

5. Is the manuscript presented in an intelligible fashion and written in standard English?

Reviewer #1: Yes

Reviewer #2: Yes

6. Review Comments to the Author

**Reviewer #1: ** The authors have addressed major concerns expressed by the reviewers by modifications to the text and title, and including new figures and references, that have been helpful in significantly improving the manuscript. With regards to a more relevant analysis using heterodimers of Zebrafish S100 proteins, the authors have offered a clear explanation for why experimentally measuring the functional properties of possible homo and heterodimeric combinations would be challenging. Their justification for addressing the issue using a pilot computational approach seems reasonable.

**Reviewer #2: ** In this revised manuscript, Orlandi and Harms provided new valuable data and experiments that helped address most of the concerns I had. Importantly, they down-tuned their interpretation about the non-existence of a functional ortholog of calprotectin, starting from a modified title, as not all calprotectin-dependent functions were tested in this study and other receptors for a calprotectin-like S100 protein in zebrafish may exist, which they also clearly stated in the results and discussion section. I also appreciate the new section on the modelling of possible homo- and hetero S100 dimers that now more firmly highlight the fact that none of them would be able to mimic the unique hexaHis site from which human calprotectin gets its antimicrobial properties. Although it would be nice to test these hypotheses experimentally, I do fully understand that this cannot be tested at this stage in the current study. I also acknowledge that the EDTA control has been added in the antimicrobial assays, as well as the test with calprotectin for NFkB activation and the inclusion of the SDS-PAGE gel for assessing calprotectin purity levels and homogeneity. These adds-on greatly ameliorated the current manuscript. Overall, I find that the manuscript has been substantially improved. I however have two remaining concerns:

1) The first one is about the analysis of the RNAseq dataset from tail fin amputation in zebrafish and the interpretation of the data made by the authors regarding the wide up-regulation of several zebrafish s100 genes in response to injury. The authors state that was is represented is the transcriptional upregulation for each gene following injury but it is unclear from which precise data these graphs were built. The published RNAseq data contains datasets from 3 distinct time points following injury (1, 2 and 4 dpa) plus datasets pre-injury. How was the analysis done to compare pre- and post-injury transcriptional profiles? Which data were employed for post-injury (only 1 time point or all 3 merged together?). This should be detailed in the experimental section as well as in the figure legend. We did analyze some of these data ourselves and actually could not find any upregulation of some of the mentioned genes following injury, therefore I find it essential that the procedure for comparison pre/post-injury is clarified.

2) I would still be carefully in stating that no functional ortholog of calprotectin for the inflammatory function of this protein are found in zebrafish. I do agree with this conclusion for the antimicrobial activity and for the infection-related pro-inflammatory function. But as previously stated, in mammals, S100A8/A9 exerts its pro-inflammatory activity in many disease contexts and through many different receptors, that can obviously not all be tested in this study. Therefore it should be more strongly emphasized in the discussion/conclusion but also in the abstract. I still don’t think that a strong conclusion can be made on the role pro-inflammatory role of zebrafish s100 proteins and whether might still have a similar function to human calprotectin in that respect.

In conclusion, while the current manuscript has been strongly improved, I would still like to see these two points addressed before it can be considered suitable for publication in PLoS ONE.

7. PLOS authors have the option to publish the peer review history of their article (what does this mean? ). If published, this will include your full peer review and any attached files.

**Do you want your identity to be public for this peer review?** For information about this choice, including consent withdrawal, please see our Privacy Policy .

Reviewer #1: **Yes: ** Geetha Srikrishna

Reviewer #2: **Yes: ** Laure Yatime

---

## [Author Response · Author response to Decision Letter 1]

28 Feb 2025

Reviewer #1: The authors have addressed major concerns expressed by the reviewers by modifications to the text and title, and including new figures and references, that have been helpful in significantly improving the manuscript. With regards to a more relevant analysis using heterodimers of Zebrafish S100 proteins, the authors have offered a clear explanation for why experimentally measuring the functional properties of possible homo and heterodimeric combinations would be challenging. Their justification for addressing the issue using a pilot computational approach seems reasonable.

Response to Reviewer 1: Thank you for helping us make important, necessary improvements to our manuscript.

Reviewer #2: In this revised manuscript, Orlandi and Harms provided new valuable data and experiments that helped address most of the concerns I had. Importantly, they down-tuned their interpretation about the non-existence of a functional ortholog of calprotectin, starting from a modified title, as not all calprotectin-dependent functions were tested in this study and other receptors for a calprotectin-like S100 protein in zebrafish may exist, which they also clearly stated in the results and discussion section. I also appreciate the new section on the modelling of possible homo- and hetero S100 dimers that now more firmly highlight the fact that none of them would be able to mimic the unique hexaHis site from which human calprotectin gets its antimicrobial properties. Although it would be nice to test these hypotheses experimentally, I do fully understand that this cannot be tested at this stage in the current study. I also acknowledge that the EDTA control has been added in the antimicrobial assays, as well as the test with calprotectin for NFkB activation and the inclusion of the SDS-PAGE gel for assessing calprotectin purity levels and homogeneity. These adds-on greatly ameliorated the current manuscript. Overall, I find that the manuscript has been substantially improved. I however have two remaining concerns:

1) The first one is about the analysis of the RNAseq dataset from tail fin amputation in zebrafish and the interpretation of the data made by the authors regarding the wide up-regulation of several zebrafish s100 genes in response to injury. The authors state that was is represented is the transcriptional upregulation for each gene following injury but it is unclear from which precise data these graphs were built. The published RNAseq data contains datasets from 3 distinct time points following injury (1, 2 and 4 dpa) plus datasets pre-injury. How was the analysis done to compare pre- and post-injury transcriptional profiles? Which data were employed for post-injury (only 1 time point or all 3 merged together?). This should be detailed in the experimental section as well as in the figure legend. We did analyze some of these data ourselves and actually could not find any upregulation of some of the mentioned genes following injury, therefore I find it essential that the procedure for comparison pre/post-injury is clarified.

2) I would still be carefully in stating that no functional ortholog of calprotectin for the inflammatory function of this protein are found in zebrafish. I do agree with this conclusion for the antimicrobial activity and for the infection-related pro-inflammatory function. But as previously stated, in mammals, S100A8/A9 exerts its pro-inflammatory activity in many disease contexts and through many different receptors, that can obviously not all be tested in this study. Therefore it should be more strongly emphasized in the discussion/conclusion but also in the abstract. I still don’t think that a strong conclusion can be made on the role pro-inflammatory role of zebrafish s100 proteins and whether might still have a similar function to human calprotectin in that respect.

In conclusion, while the current manuscript has been strongly improved, I would still like to see these two points addressed before it can be considered suitable for publication in PLoS ONE.

Response to Reviewer 2: Thank you for the feedback and for your input which greatly improved our manuscript. To address your concerns, we have made the following revisions:

1) You have made an excellent point. We did not clearly describe how we interpreted the injury dataset and, while doing this revision, we discovered that we had made some incorrect assumptions about the presentation of data on the interactive platform for this dataset. We have now included two more panels in our S3 Fig which parse the data over sample stage: pre-injury, 1-day post-amputation (dpa), 2dpa, and 4dpa. We also amended the language in the results section (lines 252-275) and S3 Figure legend (lines 1080-1091). We now assess gene expression profiles across sample timepoints and compare them to the macrophage marker mpeg1.1 expression profile. We hope this has resolved your concerns.

2) We understand your worry here since we have only tested a handful of s100 proteins to activate a singular pro-inflammatory pathway. We have made several changes to the text to address this concern, including in the abstract (line 46 and 48-49), introduction (lines 102, 106-109), and results (lines 436-439). We feel that we have sufficiently addressed this point in our discussion (lines 514-519, and 525-556). As you have reiterated, we agree we have not proven that zebrafish s100s did not convergently evolve functions similar to calprotectin. Rather, we find no evidence to support this hypothesis of convergent evolution. But we do stand by our most important claim: that zebrafish do not have an ortholog of calprotectin.

Thank you so much for all of your thoughtful comments and critiques. Our paper has improved significantly during this review process. We greatly appreciate your effort and time.

---

## [Decision Letter · Decision Letter 2]

26 Mar 2025

Zebrafish do not have a calprotectin ortholog

PONE-D-24-26499R2

Dear Dr. Orlandi,

We’re pleased to inform you that your manuscript has been judged scientifically suitable for publication and will be formally accepted for publication once it meets all outstanding technical requirements.

Kind regards,

Andrei Chernov

Academic Editor

PLOS ONE

Additional Editor Comments (optional):

Reviewers' comments:

Reviewer's Responses to Questions

**Comments to the Author**

1. If the authors have adequately addressed your comments raised in a previous round of review and you feel that this manuscript is now acceptable for publication, you may indicate that here to bypass the “Comments to the Author” section, enter your conflict of interest statement in the “Confidential to Editor” section, and submit your "Accept" recommendation.

Reviewer #2: All comments have been addressed

2. Is the manuscript technically sound, and do the data support the conclusions?

Reviewer #2: Yes

3. Has the statistical analysis been performed appropriately and rigorously? 

Reviewer #2: Yes

4. Have the authors made all data underlying the findings in their manuscript fully available?

Reviewer #2: Yes

5. Is the manuscript presented in an intelligible fashion and written in standard English?

Reviewer #2: Yes

6. Review Comments to the Author

Reviewer #2: With this revised version, the authors have addressed all the concerns I had. In particular, they have now analyzed appropriately the RNAseq data on the tail fin amputation model and provided an adequate and more understandable visualization of the results for the different s100 genes in Figure S3. I therefore recommend the current manuscript to be accepted for publication.

7. PLOS authors have the option to publish the peer review history of their article (what does this mean? ). If published, this will include your full peer review and any attached files.

**Do you want your identity to be public for this peer review?** For information about this choice, including consent withdrawal, please see our Privacy Policy .

Reviewer #2: **Yes: ** Laure Yatime

---

## [Editor Report · Acceptance letter]

PONE-D-24-26499R2

PLOS ONE

Dear Dr. Orlandi,

I'm pleased to inform you that your manuscript has been deemed suitable for publication in PLOS ONE. Congratulations! Your manuscript is now being handed over to our production team.

Kind regards,

on behalf of

Dr. Andrei Chernov

Academic Editor

PLOS ONE